# Revisiting the Scaling Properties of Downstream Metrics in Large Language Model Training

**Jakub Krajewski**[2,3*†]
gim.jakubk@gmail.com

**Amitis Shidani**[1*]
amitis_shidani@apple.com

**Dan Busbridge**[1]
dbusbridge@apple.com

**Sam Wiseman**[1]
s_wiseman@apple.com

**Jason Ramapuram**[1]
jramapuram@apple.com

[1] **Apple**     [2] **University of Warsaw**     [3] **IDEAS NCBR**

## Abstract

While scaling laws for Large Language Models (LLMs) traditionally focus on proxy metrics like pretraining loss, predicting downstream task performance has been considered unreliable. This paper challenges that view by proposing a direct framework to model the scaling of benchmark performance from the training budget. We find that for a fixed token-to-parameter ratio, a simple power law can accurately describe the scaling behavior of log accuracy on multiple popular downstream tasks. Furthermore, we introduce functional forms that predict accuracy across token-to-parameter ratios and account for inference compute under repeated sampling. Our results show that the direct approach extrapolates better than the previously proposed two-stage procedure, which is prone to compounding errors. We validate our findings on models with up to 17B parameters trained on up to 350B tokens across two dataset mixtures. To support reproducibility and encourage future research, we release the complete set of pretraining losses and downstream evaluation results.

## 1 Introduction

Large Language Models (OpenAI et al., 2024; Team et al., 2025; DeepSeek-AI et al., 2025) based on the Transformer (Vaswani et al., 2023) architecture have achieved impressive results, approaching or exceeding human-level performance across multiple domains. Scaling Laws (Hestness et al., 2017; Kaplan et al., 2020) are an established method for modeling the performance of these networks, enabling researchers to plan large-scale training runs based on curated sets of smaller experiments. Traditionally, these laws focus on predicting proxy metrics for model quality, such as pre-training log-perplexity. This has proven invaluable for optimizing training hyperparameters, like the optimal ratio of tokens to parameters.

Another important direction in understanding the scaling of LLMs is tracking the behavior of more interpretable indicators of model capabilities, like accuracy on downstream benchmarks measuring the performance on general knowledge, reasoning, math and coding tasks. Despite early attempts to solve this problem (Grattafiori et al., 2024; Isik et al., 2025; Chen et al., 2025), scaling downstream metrics have been often referred to as noisy and unreliable (Schaeffer et al., 2025b; Lourie et al., 2025).

---

[†]Work done as an intern at Apple. [*] Core contributors.

Current approaches to modeling the downstream performance performance of LLMs (Grattafiori et al., 2024; Chen et al., 2025; Bhagia et al., 2025) typically rely on a two-stage approach, where the training budget is first mapped to a proxy metric like mean log-probability of the correct answer, and then another dependence is established, mapping to benchmark accuracy.

In this paper, we propose a framework for directly predicting downstream performance from the pre-training budget. We demonstrate that when holding the token-to-parameter ratio fixed, the scaling of downstream accuracy is accurately captured by a simple power law. We validate this law across an extensive suite of 130 experiments, spanning models up to 17B parameters trained on 350B tokens, and evaluate its predictive power on twelve popular benchmarks used for evaluating LLM capabilities. Our contributions can be summarized as follows:

1. We show that the downstream model performance on multiple popular benchmarks scales predictably with respect to the pre-training budget. We train our models either using C4 or a more modern mixture dataset (Section 3) to highlight the influence of data composition.

2. We propose a *direct*, two-parameter scaling law for downstream performance. We show that, contrary to previous claims of unreliability, it accurately predicts model capabilities from the training budget and offers a simpler alternative to prior two-stage methods.

3. We extend the scaling law to model downstream performance across different token-to-parameter ratios. We further derive the formula for predicting pass rates with repeated sampling in code metrics.

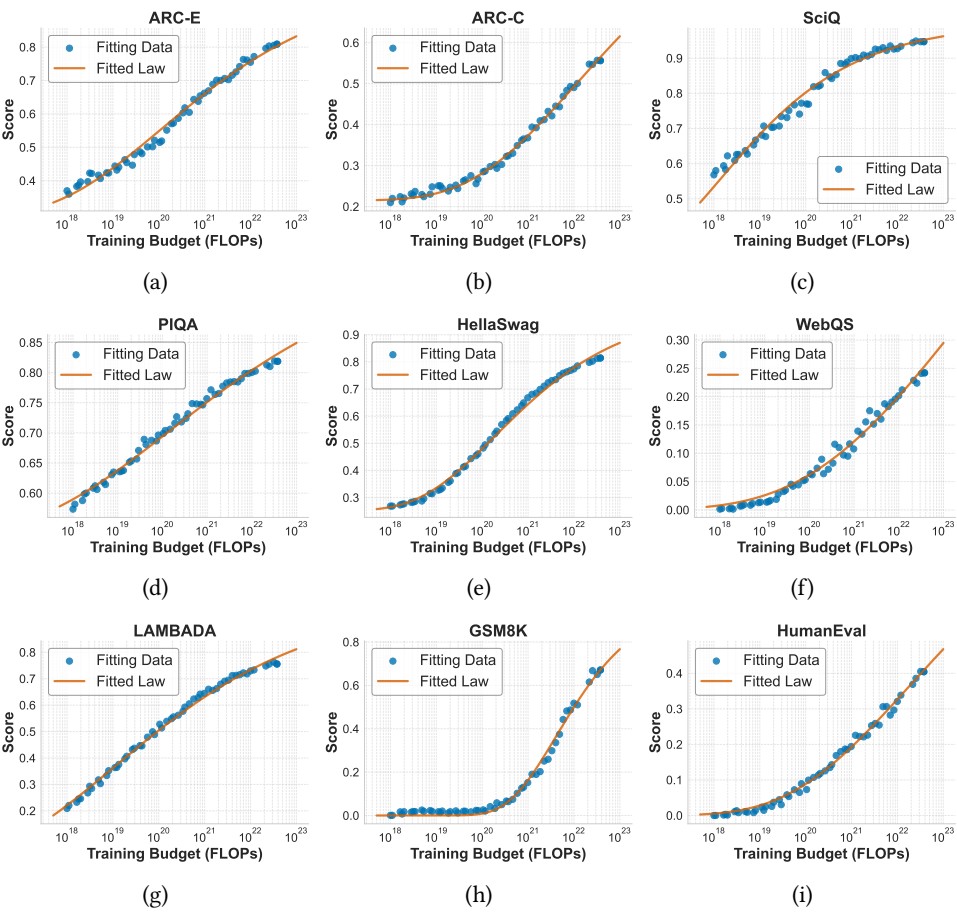

Figure 1: Benchmark accuracy can be described using a direct scaling law based on training FLOPs. The solid line represents the scaling law fit using Equation 2, and each point corresponds to accuracy measured for the final checkpoint at a given training budget.

## 2 Related Work

**Downstream scaling and what it predicts.** A large body of work proposes to forecast downstream task performance from small, cheaper experiments, but they differ in the signal one should extrapolate and in the tasks/metrics used for validation. Two-stage methods posit an intermediate proxy (typically pretraining loss or task-specific negative log-likelihood) and then map that proxy to accuracy on a benchmark. For instance, Chen et al. (2024) fit a compute→loss power law and a subsequent loss→performance map post-emergence.

LLaMA 3 (Dubey et al., 2024) adopts the same two-stage template, correlating training FLOPs with the normalized NLL of the correct choice and then using a sigmoidal link from NLL to accuracy to forecast final scores for ranked multiple-choice tasks. In parallel, loss-centric analyses formalize when and why such mappings can work: Du et al. (2024) explain emergent "breaks" through loss thresholds; Brandfonbrener et al. (2024) show that losses across datasets can be predicted from each other (loss-to-loss), offering a unifying surrogate for downstream ability. (Mayilvahanan et al., 2025) analyze the contribution of various parts of the training procedure and find the crucial importance of the dataset used. (Li et al., 2025b) provides a broad survey of different approaches and discrepancies in the scaling law literature, providing recommendations for reproducibility.

Closer to practice, Gadre et al. (2024a) report that aggregate downstream error is near-linear when plotting log-error vs. validation loss, while Bhagia et al. (2024) and Chen et al. (2024) give systematic recipes for fitting loss-to-accuracy transforms. Our study is technically aligned with these efforts in treating downstream performance as a predictable function of a small number of observable quantities, but we deliberately place the emphasis on the evaluation side: we analyze predictability across metric families (ranked classification, exact-match generation, chain-of-thought math, and code pass rates), and we stress model-agnostic procedures that require no access to internal losses beyond what the evaluation harness already.

**Compute-efficient scaling measurements.** Compute-efficient ladders (Bhagia et al., 2024) advocate measuring task scaling on a small "ladder" of models to infer the trend for a target scale, showing strong fits on ranked classification benchmarks where each item is scored by a model's log-probability over choices. By design, this restricts metrics to ranked classification and uses the model's log-likelihood of the correct answer as the proxy; it is not obvious how to choose or calibrate the proxy for two-stage prediction when the downstream metric is not ranked classification, e.g., code pass rates or exact-match generation.

DataDecide (Magnusson et al., 2025) addresses a complementary question: choosing pretraining data by running small experiments and extrapolating which data mixture will win at larger scales; methodologically, it shares our emphasis on early, low cost decisions, but it operates on the data axis rather than on evaluation predictability.

Complementing these observations, GPT-4 documents predictable scaling on evaluation tasks by fitting a power law to smaller-run models and extrapolating mean log pass rate on HumanEval by using forecasts of $10^3\times$ less compute; they also note that individual items can display non-monotonicities even when aggregate trends are smooth (OpenAI, 2023). Relative to these works, our paper focuses on forecasting evaluation outcomes directly from training FLOPs, rather than inducing possible failure modes that can arise from metric design.

**Reliability, emergence, and scope.** The literature also cautions that downstream scaling is not universally smooth: apparent "emergent abilities" can be artifacts of metric thresholds (Schaeffer et al., 2023) or true structural breaks tied to loss regimes (Du et al., 2024). These effects complicate two-stage fits that assume a single global link from proxy to accuracy (Chen et al., 2024; Bhagia et al., 2024). Our results echo this caution while aiming to be practically useful: we make conservative, metric-aware predictions, and we surface when a metric is likely to induce nonmonotonic or thresholded behavior. We view our contribution as incremental and complementary: rather than proposing a new universal law, we clarify when evaluation is predictable in practice across the metrics researchers actually use, and we provide a lightweight recipe and diagnostics that can be reproduce for a fixed set of pretraining datasets and evaluations.

## 3 Describing the Scaling Behavior of Downstream Metrics

### 3.1 Experimental Setup

We characterize the relationship between training budget and downstream performance by pre-training a comprehensive grid of models. Our experimental setup spans 48 distinct training budgets and five token-to-parameter ratios (10, 20, 40, 80, and 160), allowing us to systematically map the scaling landscape. We start by presenting the details of this setup first, and then exploring the connection between the compute and downstream performance.

**Model architecture and training hyperparameters.** We use a standard modern pre-norm decoder-only Transformer architecture. We employ the RoPE positional embedding (Su et al., 2023), and SwiGLU activation (Shazeer, 2020). We use the tokenizer from (Li et al., 2025a) with vocabulary size of 150k tokens. The sequence length is set to 4096. Further details on the training hyperparameters are provided in Appendix G.

**Dataset.** We train the models on a mixture of data covering general web data, math and code, to be able to track model capabilities in various domains. More specifically, we sample 75% of tokens from DCLM (Li et al., 2025a), 15% from Stack v2 (Lozhkov et al., 2024) (cleaned using the OpenCoder (Huang et al., 2025) heuristics and filtered to remove licenced data), and 10% from OpenMathReasoning dataset (Moshkov et al., 2025).

**Code.** Our training code is based on the open-source AXLearn repository (Lee et al., 2025).

**Evaluation benchmarks.** We consider the following benchmarks, covering various model capabilities: ARC-Easy (Clark et al., 2018), ARC-Challenge (Clark et al., 2018), SqiQ (Welbl et al., 2017), PIQA (Bisk et al., 2019), Hellaswag (Zellers et al., 2019), WebQS (Berant et al., 2013), Winogrande (Sakaguchi et al., 2019), LAMBADA (Paperno et al., 2016), TriviaQA (Joshi et al., 2017), GSM8k (Cobbe et al., 2021), HumanEval (Chen et al., 2021), LBPP (Matton et al., 2024).

### 3.2 Scaling of Accuracy with Training FLOPs

We first analyze models trained at a fixed token-to-parameter ratio (TPR) of 20, which approximates the compute-optimal point suggested by the Chinchilla scaling laws (Hoffmann et al., 2022). For this fixed TPR, we sweep across a range of compute budgets to model the relationship between downstream task performance and training FLOPs. A broader analysis, including overtrained models with different token multipliers, is presented in Section 3.3.

Modeling the scaling of downstream metrics is inherently challenging. This complexity arises because a benchmark's overall accuracy is an aggregation of scaling behaviors on individual examples, making the observed trend highly dependent on the benchmark's specific composition. Given this, we adopt an *empirical* methodology: we investigate whether a single functional form can accurately model performance scaling across a diverse set of popular downstream benchmarks.

Caballero et al. (2023) showed that a wide range of evaluation metrics can be modeled by a smooth approximation of a piecewise linear function in log-log space. We examine the fit of this function, referred to as the Broken Neural Scaling Laws (BNSL). More specifically, we model the downstream accuracy $Q$ based on the training budget $C$ using the following equation:

$$Q = a + b\, C^{-c_0} \left(1 + \left(\frac{C}{d_1}\right)^{1/f_1}\right)^{-c_1 f_1}, \tag{1}$$

which is a BNSL with one transition point. We fit the parameters $a, c_0, d_1, f_1$ based on data.

We further examine the possibility of modeling the curves using a simplified formula with fewer parameters. The first natural idea would be to eliminate the breaking points from Equation 1, resulting in a simple power law. However, such a functional form would be strictly concave as a function of $C$, an assumption not justified by the observed data. For example, ARC-Easy and HellaSwag in Figure 1 are convex in the lower compute range and concave for higher budgets.

Therefore, we consider modeling the *log accuracy* as a power law, a functional form that has the ability of describing the observed $S-$shaped behavior. In the preliminary exploration, we also noticed an approximately linear relationship between log accuracy and training FLOPs on a log-log scale. More precisely, we consider the following functional form:

$$-\log(Q) = \frac{A}{C^\alpha}, \tag{2}$$

where $A > 0$ and $\alpha > 0$ are benchmark-specific coefficients fitted on the data.

**Lower asymptote for Accuracy.** Note that Equation 2 indicates that $Q$ takes values from a fixed range, with the lower and upper asymptotes set to zero and one, respectively. This cannot accurately describe the behavior of multiple-choice benchmarks, where even choosing random option results in $Q > 0$. Therefore, for such tasks, we normalize accuracy to take values from the interval $[0, 1]$. More specifically, we first apply the transformation

$$Q' := (Q - Q_{\text{random}})/(1 - Q_{\text{random}}), \tag{3}$$

where $Q_{\text{random}}$ is the random accuracy for a given benchmark and metric type. For multiple choice tasks, we estimate $Q_{\text{random}}$ by training a set of 10 small models in the compute scale below $1 \times 10^{17}$ FLOPs and calculate the average of their performance.

**Fitting and validation.** We empirically confirm the validity of the functional form of Equation 2 by examining the quality of the fit presented in Figure 1. We further evaluate our scaling law by fitting coefficients to experiments with the training budget below $6 \times 10^{21}$ FLOPs, leaving the remaining runs as the validation set. The aggregated absolute and relative errors averaged across the metrics are shown in Table 1.

Table 1: Fit quality of the power law functional form.

| Train MAE | Valid MAE | Train MRE (%) | Valid MRE (%) |
|-----------|-----------|---------------|---------------|
| 0.0086    | 0.0195    | 0.86%         | 1.95%         |

### 3.3 EXTENDING ACROSS TOKEN-TO-PARAMETER RATIOS

In this section, we extend the previously proposed results, which were dependent solely on training compute, to different token-to-parameter ratios.

Our model for log accuracy scaling (Equation 2) parallels the standard power laws used for log perplexity (Henighan et al., 2020; Hoffmann et al., 2022). A key distinction is the absence of an irreducible term in our model. In perplexity scaling, this term accounts for the inherent entropy of the data, creating a performance floor. For accuracy, however, the theoretical performance ceiling is one. We therefore assume that with an infinite compute budget, a model can achieve perfect accuracy, making an irreducible error term unnecessary.

Building on this analogy, we model the negative log accuracy, as a function of model parameters $N$ and dataset size $D$. We adapt the functional form for pretraining loss from Hoffmann et al. (2022). However, consistent with our earlier reasoning, we exclude the irreducible error term. We fit the coefficients $A, \alpha, B$, and $\beta$ in the resulting equation:

$$-\log Q = \frac{A}{N^\alpha} + \frac{B}{D^\beta}. \tag{4}$$

**Fitting and validation.** We fit the coefficients of Equation 4 by minimizing the Huber loss ($\delta = 1e - 3$) with the L-BFGS-B algorithm. To evaluate the model's predictive power, we established a hold-out validation set comprising all runs with either a training budget over $6 \times 10^{21}$ FLOPs or a token-to-parameter ratio (TPR) greater than 80. The model coefficients were then fit exclusively on the remaining training data. Table 2 reports the absolute and relative errors on this validation set, demonstrating a strong predictive fit.

Figure 2 provides a visual confirmation of the model's quality. For the Lambada benchmark, panels (a)-(c) show the close alignment between observed data and the fitted curves across different TPRs. Panels (d)-(f) further illustrate the model's effectiveness on three additional benchmarks.

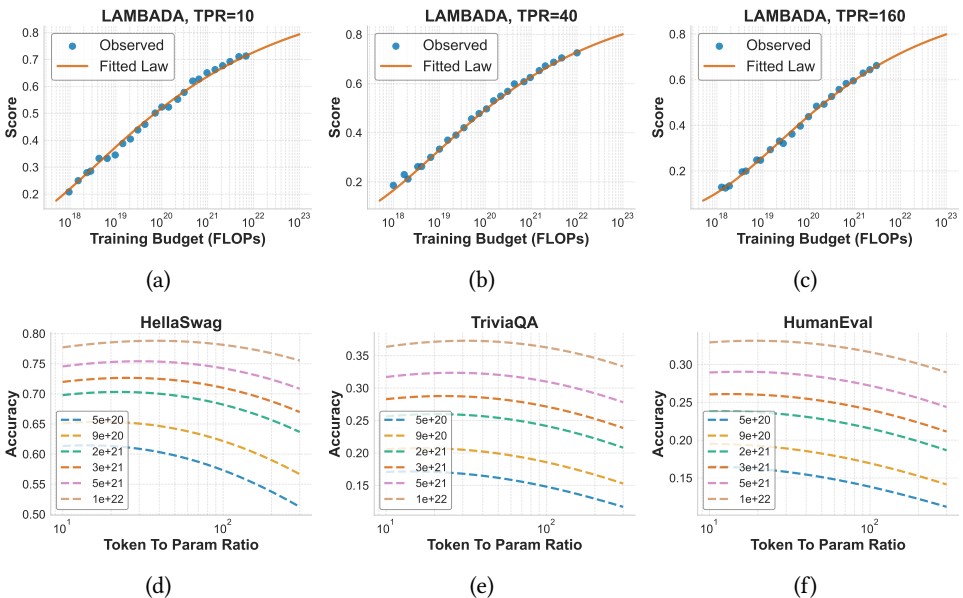

Figure 2: Comparison of scaling the downstream accuracy on different token-to-parameter ratios.

Table 2: Fit quality of Equation 4.

| Train MAE | Valid MAE | Train MRE (%) | Valid MRE (%) |
|-----------|-----------|---------------|---------------|
| 0.0103 | 0.0191 | 1.03% | 1.91% |

## 3.4 Considering the Effect of Repeated Sampling with Inference Compute

We also study the effect of increasing the number of samples in pass@k for coding benchmarks across different training budgets.

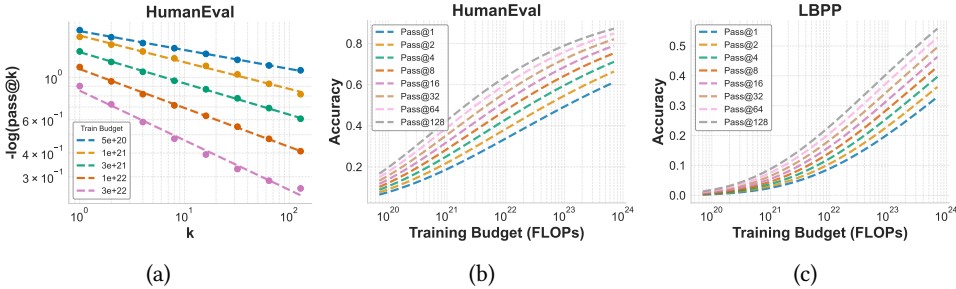

Figure 3: Comparison of pass@k behaviour across tasks. (a) Intuition for the functional form. (b) Predicted pass rate curves for HumanEval. (c) Predicted pass rate curves for LBPP.

Figure 3 plots the negative log pass rate against k for the HumanEval benchmark. The plot highlights two primary observations:

1. For a fixed training budget, the relationship is approximately linear in log scale, indicating that the rate follows a power law with respect to $k$. This observation is consistent with findings from related work (Brown et al., 2024; Hughes et al., 2024; Kwok et al., 2025), observing power law scaling with the number of attempts $k$ for coding, math, and adversarial robustness.

2. The slope of this linear relationship depends on the training compute budget, becoming steeper for larger compute. Concurrently to this work, similar finding is presented in (Schaeffer et al., 2025b).

Connecting these observations with scaling law in Equation 2 for $k = 1$, we propose the following equation for modeling the pass@k rate $Q$ based on the training budget $C$ and number of trials $k$:

$$\log(-\log Q(C, k)) = \log A + \alpha \log C + \beta \log k + \delta \log C \log k \quad (5)$$

**Fitting and validation.** We fit the scaling law in Equation 5 to HumanEval pass@k data (see Figure 3 for visualizations). For validation, we established a hold-out set, fitting the model's coefficients on experiments with FLOPs below $6 \times 10^{21}$ and $k \leq 32$. The model's absolute and relative errors on the remaining validation data, reported in Table 3, confirm its predictive accuracy.

Table 3: Fit quality of Equation 5.

| Train MAE | Valid MAE | Train MRE (%) | Valid MRE (%) |
|-----------|-----------|---------------|---------------|
| 0.0111 | 0.0284 | 6.30% | 7.94% |

**Theoretical justification.** In Appendix C we derive analytical upper and lower bounds for the pass rate given $k$ independent trials. These results can serve as complementary for deriving the functional form of Equation 5. Theoretical analysis of the mechanisms behind power law scaling of repeated sampling is also explored in related work (Schaeffer et al., 2025a; Levi, 2024; Kazdan et al., 2025).

## 4 Predicting Accuracy of Large Models

### 4.1 Directly Predicting Accuracy from Training Compute

We leverage the power-law scaling law from Section 3.2 to extrapolate model performance from smaller to larger compute budgets. To test the aproach, we fit the scaling law coefficients using only experiments trained with $3 \times 10^{18}$ to $6 \times 10^{21}$ FLOPs. When fitting coefficients in Equation 2, we only use runs with accuracy at least 5% points above the random performance, as we notice that these small accuracy results tend to have larger variance disturbing the fit. We then use this model to predict the accuracy for larger, held-out models, whose training budget is up to 6.7x larger. For comparison, we also fit the coefficients in BNSL (Equation (1)) and assess the prediction accuracy on the same validation set of experiments. The results are presented in Table 4. We notice that while both functional forms achieve a good extrapolation quality (below 3% points MAE on average), our proposed power law scaling achieves a slightly lower validation error.

Table 4: MRE and MAE in downstream accuracy prediction for BNSL and Power Law scaling strategies.

| Benchmark | Metric Type | Power Law | | BNSL | |
|-----------|-------------|-----------|---------|----------|---------|
| | | MAE | MRE (%) | MAE | MRE (%) |
| ARC-E | acc_norm | 0.0186 | 2.37 | 0.0555 | 7.00 |
| ARC-C | acc_norm | 0.0068 | 1.30 | 0.0155 | 2.86 |
| SciQ | acc_norm | 0.0051 | 0.55 | 0.0045 | 0.49 |
| PIQA | acc_norm | 0.0151 | 1.86 | 0.0087 | 1.06 |
| HellaSwag | acc_norm | 0.0298 | 3.74 | 0.0173 | 2.15 |
| Winogrande | acc | 0.0119 | 1.64 | 0.0363 | 4.93 |
| WebQS | exact_match | 0.0166 | 7.28 | 0.0033 | 1.46 |
| TriviaQA | exact_match | 0.0259 | 6.17 | 0.0424 | 10.13 |
| LAMBADA | acc | 0.0258 | 3.45 | 0.0043 | 0.58 |
| GSM8K | exact_match | 0.0559 | 9.67 | 0.0807 | 13.90 |
| HumanEval | pass@1 | 0.0171 | 5.11 | 0.0293 | 8.29 |
| LBPP | pass@1 | 0.0154 | 13.43 | 0.0066 | 6.70 |
| Average | | 0.0203 | 4.72 | 0.0254 | 4.96 |

## 4.2 Predicting Downstream Accuracy in a Two-Stage Approach

Previous works (Bhagia et al., 2025; Chen et al., 2025) use a *two-stage* approach to predict downstream task performance based on model characteristics such as FLOPs, size, and training data. The intuition behind this approach is that the model's loss acts as a proxy for predicting downstream task accuracy. However, we argue that while this approach provides useful interpretability in some cases, the multi-stage nature of this method compounds errors from each stage, ultimately resulting in scaling laws with higher variance and reduced predictive accuracy.

To demonstrate this, we explore the correlation between various proxy metrics, such as log-probability, evaluation loss, Brier Score, and final accuracy on downstream benchmarks. We then implement the two-stage approach on our dataset and compare its performance against our proposed direct scaling law, which directly predicts downstream accuracy from FLOPs.

### 4.2.1 Correlation Between Proxy Metrics and Downstream Accuracy

The two-stage framework relies on the critical assumption that an intermediate proxy metric, such as a model's training loss, strongly correlates with its final downstream accuracy. In this section, we analyze the correlation between downstream task accuracy and several candidate proxy metrics including log-probability, evaluation loss, and the Brier Score. A strong and consistent correlation is a prerequisite for the two-stage method to be effective, as any poor choice in this relationship will propagate errors and reduce the accuracy of the final scaling predictions.

To capture the potentially non-linear dependency between proxy metrics and downstream task performance, we fit logistic functions of the form Acc $= 1/(1 + e^{-a \cdot \text{proxy} + b})$ and compare the goodness of fit across metrics. We evaluate fit quality using Root Mean Squared Error (RMSE) and the coefficient of determination ($R^2$). Note that the choice of proxy metric does not affect the first stage of the two-stage approach, as Brandfonbrener et al. (2024) demonstrated that all metrics exhibit similar scaling behavior with respect to model size and compute.

Our analysis shows that most proxy metrics demonstrate strong predictive power for downstream task performance, with goodness-of-fit statistics falling within a similar range across all metrics ($R^2 > 0.95$). This suggests that practitioners can select proxy metrics based on computational convenience or availability rather than predictive accuracy. Figure 4 shows a representative example for the `arc_easy` benchmark; we observe comparable correlation strength across other benchmarks as well (see Appendix D for comprehensive results and detailed fit statistics for each proxy metric). These findings validate the assumption underlying the two-stage framework and demonstrate that the choice of proxy metric is unlikely to be a limiting factor in prediction accuracy.

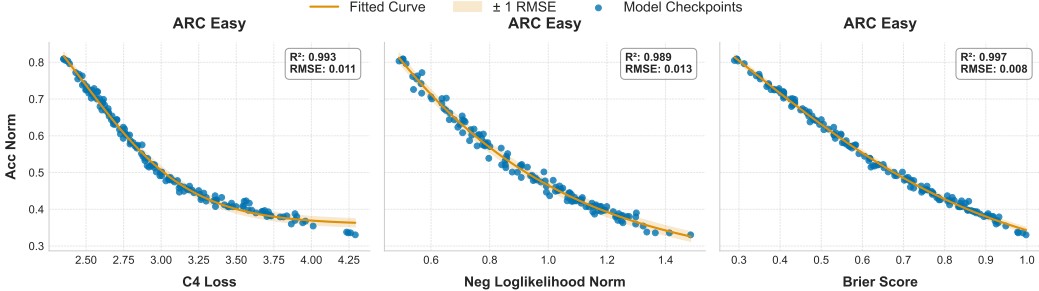

Figure 4: Dependency between downstream task and proxy metric candidates. All metrics demonstrate strong prediction power, i.e. high $R^2$ and low RMSE.

## 4.3 Two-Stage Approach is not as Strong as Direct Approach

### 4.3.1 Two-Stage Approach is not as Strong as Direct Approach

Following prior work Bhagia et al. (2025); Chen et al. (2025), we evaluate two-stage scaling laws that use negative log-likelihood (NLL) as a proxy metric to predict downstream task performance. This approach is based on the hypothesis that language modeling capability correlates with task-

specific accuracy. As established in the previous section, our analysis showed that the choice of proxy metric does not significantly impact the final prediction; therefore, we adopt NLL to maintain consistency with these baselines.

The two-stage methods map the proxy metric, $L$, to accuracy using two transition formulas: (1) a *linear* transition, Acc $= a + bL$, and (2) a *logistic* transition, Acc $= [a/(1 + e^{-k(L-L_0)})] + b$, where $a$, $b$, $k$, and $L_0$ are fitted parameters. We implement both formulas and compare them against our direct scaling approach. For all models, we fit the scaling laws on data up to a compute budget of $6 \times 10^{21}$ FLOPs and validate on models trained with greater compute.

To evaluate predictive performance, we use Mean Relative Error (MRE) and Mean Absolute Error (MAE). To assess goodness-of-fit on the training data, we use RMSE and $R^2$. These metrics, averaged over all benchmarks, are reported in Table 5. An illustrative comparison of the fits is shown in Figure 5, with additional examples in Section D, Figure 7.

Our findings indicate that the direct approaches (BNSL and the simple scaling law) consistently outperform the two-stage methods in prediction. This holds true even though the two-stage models often exhibit a superior goodness-of-fit (higher $R^2$ and lower RMSE) on the training data. For example, Figure 5 shows the two-stage logistic model achieving a better fit than BNSL, yet failing to extrapolate accurately. We attribute this discrepancy to the *compounding of errors*: inaccuracies in the first stage (FLOPs-to-NLL) are amplified by the second stage (NLL-to-Accuracy), leading to poor overall predictive power compare to the direct approach.

Moreover, we conduct an additional analysis to examine the effect of the FLOPs threshold on the quality of scaling fits. This can be interpreted as a form of sensitivity analysis: we ask what minimum FLOPs threshold is required for each benchmark and scaling law to achieve a reliable fit with an error below a given rate. The results are reported in Section E. In summary, we find that BNSL and the simple power law exhibit relatively stable behaviour across a wide range of FLOPs, whereas the two-stage linear and logistic models are less robust and often yield higher error rates.

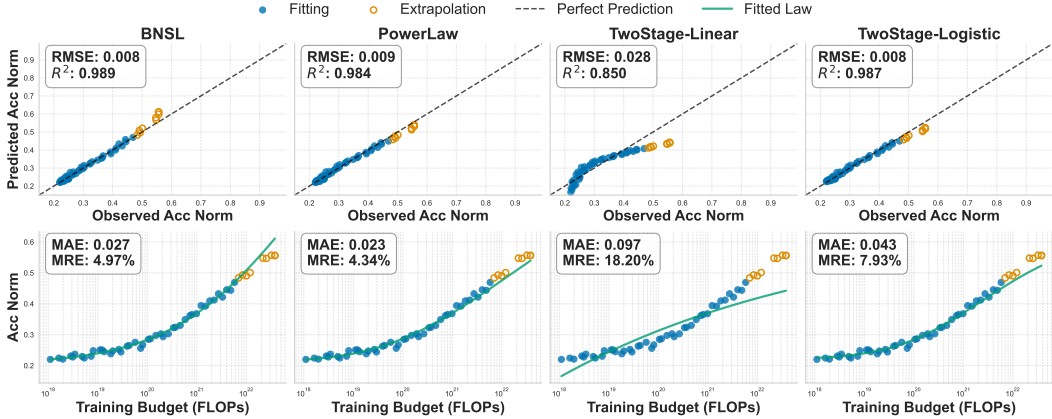

Figure 5: Scaling law fits. Comparing the direct approaches (BNSL, simple power law) from Section 3.2 with two-stage approaches (Linear and Logistic) for ARC Challenge.

Table 5: Performance comparison of scaling law strategies across different error metrics.

| Scaling Law Strategy | MRE (%) | | MAE | | RMSE | | R² | |
|---|---|---|---|---|---|---|---|---|
| | mean | std | mean | std | mean | std | mean | std |
| PowerLaw | 1.963 | 1.201 | 0.015 | 0.010 | 0.011 | 0.006 | 0.986 | 0.011 |
| BNSL | 2.713 | 2.569 | 0.020 | 0.020 | 0.007 | 0.003 | 0.993 | 0.004 |
| TwoStage-Linear | 6.667 | 6.958 | 0.044 | 0.034 | 0.023 | 0.011 | 0.943 | 0.054 |
| TwoStage-Logistic | 6.351 | 3.278 | 0.047 | 0.024 | 0.017 | 0.006 | 0.974 | 0.013 |

## 5    Discussion

### 5.1    Effect of the Data Mixture

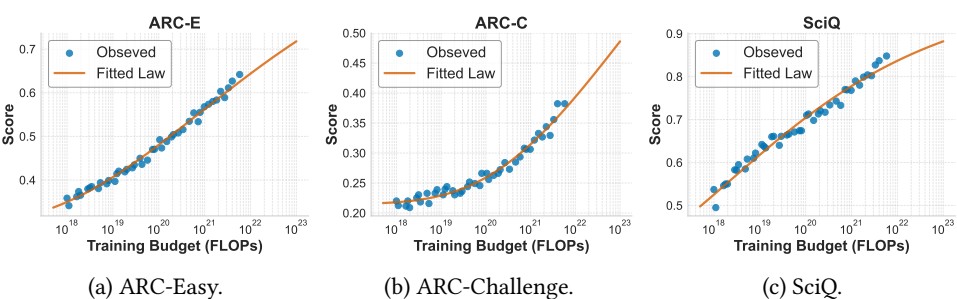

(a) ARC-Easy.      (b) ARC-Challenge.      (c) SciQ.

Figure 6: Example downstream scaling curves when changing the pre-training dataset to C4. All metrics are shown in Appendix F.

A natural question to consider is whether the previously presented scaling curves are specific to one particular data mixture. To address this concern, we train a suite of models with the same setup as described in Section 3, but change the dataset to C4 (Raffel et al., 2023). We fix the token-to-parameter ratio to 20 for these experiments and consider 44 training budgets between $1 \times 10^{18}$ and $6 \times 10^{21}$ FLOPs. We note random performance of these models on code and math specific downstream tasks, due to the lack of alignment of the dataset towards these domains. Overall, we observe accuracy of more than 10 percentage points above random chance on 8 downstream tasks. We fit coefficients in Equation (2) to each of them and present example plots in Figure 10. We observe good fit quality, similar to observed in Section 3, indicating that the validity of the presented scaling trends are not restricted to one particular data mixture.

## 6    Conclusions

In this work, we demonstrate that downstream benchmark accuracy scales predictably with training compute. We introduce a simple, direct scaling law that accurately models the relationship between training FLOPs and final benchmark performance. By establishing this predictable scaling behavior, our work helps make the development of large-scale models more systematic and efficient, presenting the scaling of downstream capabilities as a direct and measurable consequence of scale.

## 7    Reproducibility

We aim to support reproducibility and encourage future research based on the results of this work. We detail training hyperparameters and model training details in Section 3.1 and Appendix G. All of the models were trained using a codebase based on the open-source AXLearn repository, using publicly available datasets. We release complete set of pretraining losses and evaluation results of the models used in this study, and scaling law fitting code in a repository available under the link: https://github.com/apple/ml-scaling-downstream-metrics.

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

# Appendices

## A    Limitations and Future Work

Our claims are empirical and conditional. We model downstream accuracy as a simple function of training compute (Equation 2) and extend to parameters–tokens (Equation 4) and pass@k (Equation 5), but we do not yet offer a mechanistic account of these forms; connecting them to item-difficulty mixtures or error-decay processes remains open.

- **Metric Dependence.** Aggregate trends are smooth on the benchmarks we study, but composite or thresholded tasks can look step-like or non-monotone (e.g., BIG-bench–style mixes). A principled "metric audit" that partitions by latent difficulty would clarify when predictability should be expected.

- **Data Mixture.** The composition of the pretraining data significantly affects scaling behavior. For example, a C4-only model preserves scaling on general QA while its performance on code and math reverts toward chance (Figure 10). Scaling claims should therefore be understood as conditional on the specific data mixture and filtering pipeline used.

- **Scale Thresholds.** Reliable extrapolation appears only after task-specific FLOPs thresholds; below them, fits can be brittle (Figures 8 and 9). Future protocols should declare the threshold used and report success probabilities vs. that threshold.

- **Training Recipe Scope.** Results are from decoder-only Transformers with a fixed modern recipe; we did not test MoE, retrieval, or multimodal models, nor alternative optimizers/schedules. Exponents may be recipe-dependent.

- **Beyond Pretraining.** We fit laws on pretraining checkpoints; continued pretraining, instruction tuning, and preference optimization can reshape downstream accuracy and may require extended forms or new covariates.

- **Uncertainty and compute trade-offs** We emphasize mean errors but not calibrated prediction intervals; bootstrap-based intervals and floor estimation (Equation 3) would aid decision-making. For code, train vs. inference compute (pass@k) trade-offs deserve explicit Pareto analyses.

In summary, our framework helps reconcile predictable and unpredictable scaling phenomena. Predictability emerges beyond a task and metric specific scale threshold and under a fixed data mixture. Making these preconditions explicit should improve reproducibility across studies.

## B    Experiment List

Below we present a full list of models trained as a part of this study. In the *datasets* column, DCLM+code+math indicates the mixture described in Section 3. A subset of models was also trained on the C4 dataset, as part of the ablation in Section 5.1.

Table 6: List of experiments (Part 1/3).

| params | flops | tokens | layers | hidden dim | heads | datasets |
|--------|-------|--------|--------|-----------|-------|----------|
| 17.58B | 3.77e+22 | 357.4B | 44 | 5632 | 44 | DCLM+code+math |
| 17.58B | 3.77e+22 | 357.4B | 44 | 5632 | 44 | DCLM+code+math |
| 17.58B | 3.77e+22 | 357.4B | 44 | 5632 | 44 | DCLM+code+math |
| 16.51B | 3.22e+22 | 324.7B | 43 | 5504 | 43 | DCLM+code+math |
| 14.33B | 2.54e+22 | 295.0B | 41 | 5248 | 41 | DCLM+code+math |
| 13.40B | 2.15e+22 | 268.0B | 40 | 5120 | 40 | DCLM+code+math |
| 10.71B | 6.85e+21 | 106.6B | 37 | 4736 | 37 | DCLM+code+math |
| 9.88B | 1.19e+22 | 201.0B | 36 | 4608 | 36 | DCLM+code+math |
| 9.10B | 9.97e+21 | 182.6B | 35 | 4480 | 35 | DCLM+code+math |
| 9.10B | 4.80e+21 | 88.0B | 35 | 4480 | 35 | DCLM+code+math |
| 8.41B | 8.37e+21 | 165.9B | 34 | 4352 | 34 | DCLM+code+math |
| 7.71B | 6.98e+21 | 150.7B | 33 | 4224 | 33 | DCLM+code+math |
| 7.05B | 3.07e+21 | 72.6B | 32 | 4096 | 32 | DCLM+code+math |
| 7.05B | 5.80e+21 | 136.9B | 32 | 4096 | 32 | DCLM+code+math, C4 |
| 6.48B | 4.84e+21 | 124.4B | 31 | 3968 | 31 | DCLM+code+math |
| 6.48B | 1.00e+22 | 258.2B | 31 | 3968 | 31 | DCLM+code+math |
| 5.90B | 4.00e+21 | 113.0B | 30 | 3840 | 30 | DCLM+code+math, C4 |
| 5.90B | 2.12e+21 | 59.9B | 30 | 3840 | 30 | DCLM+code+math |
| 5.35B | 3.30e+21 | 102.7B | 29 | 3712 | 29 | DCLM+code+math, C4 |
| 4.88B | 1.45e+21 | 49.5B | 28 | 3584 | 28 | DCLM+code+math |
| 4.88B | 2.73e+21 | 93.3B | 28 | 3584 | 28 | DCLM+code+math, C4 |
| 4.40B | 4.64e+21 | 175.9B | 27 | 3456 | 27 | DCLM+code+math |
| 4.40B | 2.24e+21 | 84.7B | 27 | 3456 | 27 | DCLM+code+math, C4 |
| 3.96B | 7.15e+21 | 301.4B | 26 | 3328 | 26 | DCLM+code+math |
| 3.96B | 9.69e+20 | 40.8B | 26 | 3328 | 26 | DCLM+code+math |
| 3.96B | 1.83e+21 | 77.0B | 26 | 3328 | 26 | DCLM+code+math, C4 |
| 3.57B | 1.50e+21 | 69.9B | 25 | 3200 | 25 | DCLM+code+math, C4 |
| 3.57B | 3.11e+21 | 145.2B | 25 | 3200 | 25 | DCLM+code+math |
| 3.19B | 6.45e+20 | 33.7B | 24 | 3072 | 24 | DCLM+code+math |
| 3.19B | 1.22e+21 | 63.5B | 24 | 3072 | 24 | DCLM+code+math, C4 |
| 3.19B | 4.76e+21 | 248.8B | 24 | 3072 | 24 | DCLM+code+math |
| 2.84B | 9.82e+20 | 57.7B | 23 | 2944 | 23 | DCLM+code+math, C4 |
| 2.84B | 2.04e+21 | 119.8B | 23 | 2944 | 23 | DCLM+code+math |
| 2.84B | 4.73e+20 | 27.8B | 23 | 2944 | 23 | DCLM+code+math |
| 2.53B | 7.25e+20 | 47.7B | 22 | 2816 | 22 | DCLM+code+math, C4 |
| 2.53B | 1.50e+21 | 98.9B | 22 | 2816 | 22 | DCLM+code+math |
| 2.53B | 3.12e+21 | 205.3B | 22 | 2816 | 22 | DCLM+code+math |
| 2.53B | 7.97e+20 | 52.5B | 22 | 2816 | 22 | DCLM+code+math, C4 |
| 2.23B | 5.80e+20 | 43.3B | 21 | 2688 | 21 | DCLM+code+math, C4 |
| 2.23B | 2.27e+21 | 169.5B | 21 | 2688 | 21 | DCLM+code+math |
| 2.23B | 3.08e+20 | 23.0B | 21 | 2688 | 21 | DCLM+code+math |
| 1.96B | 9.60e+20 | 81.6B | 20 | 2560 | 20 | DCLM+code+math |
| 1.96B | 4.62e+20 | 39.3B | 20 | 2560 | 20 | DCLM+code+math, C4 |
| 1.96B | 2.23e+20 | 18.9B | 20 | 2560 | 20 | DCLM+code+math |
| 1.73B | 6.99e+20 | 67.4B | 19 | 2432 | 19 | DCLM+code+math |
| 1.73B | 3.71e+20 | 35.7B | 19 | 2432 | 19 | DCLM+code+math, C4 |

Table 7: List of experiments (Part 2/3).

| params | flops | tokens | layers | hidden dim | heads | datasets |
|--------|-------|--------|--------|------------|-------|----------|
| 1.73B | 1.45e+21 | 139.9B | 19 | 2432 | 19 | DCLM+code+math |
| 1.73B | 3.01e+21 | 290.4B | 19 | 2432 | 19 | DCLM+code+math |
| 1.73B | 3.37e+20 | 32.5B | 19 | 2432 | 19 | DCLM+code+math, C4 |
| 1.50B | 2.66e+20 | 29.5B | 18 | 2304 | 18 | DCLM+code+math, C4 |
| 1.50B | 2.16e+21 | 239.7B | 18 | 2304 | 18 | DCLM+code+math |
| 1.50B | 1.41e+20 | 15.6B | 18 | 2304 | 18 | DCLM+code+math |
| 1.50B | 1.04e+21 | 115.5B | 18 | 2304 | 18 | DCLM+code+math |
| 1.30B | 1.00e+20 | 12.9B | 17 | 2176 | 17 | DCLM+code+math |
| 1.30B | 2.08e+20 | 26.8B | 17 | 2176 | 17 | DCLM+code+math, C4 |
| 1.30B | 1.54e+21 | 197.8B | 17 | 2176 | 17 | DCLM+code+math |
| 1.30B | 1.89e+20 | 24.3B | 17 | 2176 | 17 | DCLM+code+math, C4 |
| 1.30B | 4.32e+20 | 55.6B | 17 | 2176 | 17 | DCLM+code+math |
| 1.12B | 6.43e+20 | 95.3B | 16 | 2048 | 16 | DCLM+code+math |
| 1.12B | 3.10e+20 | 45.9B | 16 | 2048 | 16 | DCLM+code+math |
| 1.12B | 1.49e+20 | 22.1B | 16 | 2048 | 16 | DCLM+code+math, C4 |
| 1.12B | 7.19e+19 | 10.7B | 16 | 2048 | 16 | DCLM+code+math |
| 0.96B | 2.18e+20 | 37.9B | 15 | 1920 | 15 | DCLM+code+math |
| 0.96B | 4.52e+20 | 78.7B | 15 | 1920 | 15 | DCLM+code+math |
| 0.96B | 1.16e+20 | 20.1B | 15 | 1920 | 15 | DCLM+code+math, C4 |
| 0.96B | 9.39e+20 | 163.3B | 15 | 1920 | 15 | DCLM+code+math |
| 0.96B | 1.05e+20 | 18.3B | 15 | 1920 | 15 | DCLM+code+math, C4 |
| 0.81B | 1.52e+20 | 31.3B | 14 | 1792 | 14 | DCLM+code+math |
| 0.81B | 4.28e+19 | 8.8B | 14 | 1792 | 14 | DCLM+code+math |
| 0.81B | 6.56e+20 | 134.8B | 14 | 1792 | 14 | DCLM+code+math |
| 0.81B | 3.16e+20 | 64.9B | 14 | 1792 | 14 | DCLM+code+math |
| 0.81B | 8.08e+19 | 16.6B | 14 | 1792 | 14 | DCLM+code+math, C4 |
| 0.81B | 7.34e+19 | 15.1B | 14 | 1792 | 14 | DCLM+code+math, C4 |
| 0.69B | 2.22e+20 | 53.6B | 13 | 1664 | 13 | DCLM+code+math |
| 0.69B | 5.67e+19 | 13.7B | 13 | 1664 | 13 | DCLM+code+math, C4 |
| 0.69B | 1.07e+20 | 25.8B | 13 | 1664 | 13 | DCLM+code+math |
| 0.69B | 4.61e+20 | 111.3B | 13 | 1664 | 13 | DCLM+code+math |
| 0.69B | 3.01e+19 | 7.3B | 13 | 1664 | 13 | DCLM+code+math |
| 0.58B | 1.53e+20 | 44.2B | 12 | 1536 | 12 | DCLM+code+math |
| 0.58B | 2.07e+19 | 6.0B | 12 | 1536 | 12 | DCLM+code+math |
| 0.58B | 4.30e+19 | 12.4B | 12 | 1536 | 12 | DCLM+code+math, C4 |
| 0.58B | 3.90e+19 | 11.3B | 12 | 1536 | 12 | DCLM+code+math, C4 |
| 0.58B | 3.17e+20 | 91.8B | 12 | 1536 | 12 | DCLM+code+math |
| 0.58B | 7.36e+19 | 21.3B | 12 | 1536 | 12 | DCLM+code+math |
| 0.48B | 5.02e+19 | 17.6B | 11 | 1408 | 11 | DCLM+code+math |
| 0.48B | 1.04e+20 | 36.5B | 11 | 1408 | 11 | DCLM+code+math |
| 0.48B | 1.41e+19 | 4.9B | 11 | 1408 | 11 | DCLM+code+math |
| 0.48B | 2.16e+20 | 75.8B | 11 | 1408 | 11 | DCLM+code+math |
| 0.48B | 2.93e+19 | 10.3B | 11 | 1408 | 11 | DCLM+code+math, C4 |
| 0.48B | 2.66e+19 | 9.3B | 11 | 1408 | 11 | DCLM+code+math, C4 |
| 0.39B | 3.44e+19 | 14.5B | 10 | 1280 | 10 | DCLM+code+math |
| 0.39B | 1.48e+20 | 62.6B | 10 | 1280 | 10 | DCLM+code+math |
| 0.39B | 9.67e+18 | 4.1B | 10 | 1280 | 10 | DCLM+code+math |
| 0.39B | 7.14e+19 | 30.1B | 10 | 1280 | 10 | DCLM+code+math |
| 0.39B | 2.01e+19 | 8.5B | 10 | 1280 | 10 | DCLM+code+math, C4 |
| 0.39B | 1.82e+19 | 7.7B | 10 | 1280 | 10 | DCLM+code+math, C4 |
| 0.32B | 6.48e+18 | 3.4B | 9 | 1152 | 9 | DCLM+code+math |

Table 8: List of experiments (Part 3/3).

| params | flops | tokens | layers | hidden dim | heads | datasets |
|---|---|---|---|---|---|---|
| 0.32B | 1.22e+19 | 6.4B | 9 | 1152 | 9 | DCLM+code+math, C4 |
| 0.32B | 1.34e+19 | 7.0B | 9 | 1152 | 9 | DCLM+code+math, C4 |
| 0.32B | 1.11e+19 | 5.8B | 9 | 1152 | 9 | DCLM+code+math, C4 |
| 0.32B | 9.92e+19 | 51.6B | 9 | 1152 | 9 | DCLM+code+math |
| 0.32B | 4.78e+19 | 24.9B | 9 | 1152 | 9 | DCLM+code+math |
| 0.32B | 2.30e+19 | 12.0B | 9 | 1152 | 9 | DCLM+code+math |
| 0.26B | 4.29e+18 | 2.8B | 8 | 1024 | 8 | DCLM+code+math |
| 0.26B | 1.52e+19 | 9.9B | 8 | 1024 | 8 | DCLM+code+math |
| 0.26B | 6.57e+19 | 42.6B | 8 | 1024 | 8 | DCLM+code+math |
| 0.26B | 8.08e+18 | 5.2B | 8 | 1024 | 8 | DCLM+code+math, C4 |
| 0.26B | 3.17e+19 | 20.5B | 8 | 1024 | 8 | DCLM+code+math |
| 0.26B | 7.35e+18 | 4.8B | 8 | 1024 | 8 | DCLM+code+math, C4 |
| 0.21B | 1.01e+19 | 8.2B | 7 | 896 | 7 | DCLM+code+math |
| 0.21B | 4.84e+18 | 3.9B | 7 | 896 | 7 | DCLM+code+math, C4 |
| 0.21B | 2.83e+18 | 2.3B | 7 | 896 | 7 | DCLM+code+math |
| 0.21B | 5.33e+18 | 4.3B | 7 | 896 | 7 | DCLM+code+math, C4 |
| 0.21B | 2.09e+19 | 16.9B | 7 | 896 | 7 | DCLM+code+math |
| 0.21B | 4.33e+19 | 35.2B | 7 | 896 | 7 | DCLM+code+math |
| 0.21B | 2.33e+18 | 1.9B | 7 | 896 | 7 | DCLM+code+math |
| 0.16B | 2.31e+19 | 24.0B | 6 | 768 | 6 | DCLM+code+math |
| 0.16B | 1.11e+19 | 11.5B | 6 | 768 | 6 | DCLM+code+math |
| 0.16B | 2.80e+19 | 29.0B | 6 | 768 | 6 | DCLM+code+math |
| 0.16B | 1.35e+19 | 14.0B | 6 | 768 | 6 | DCLM+code+math |
| 0.16B | 2.84e+18 | 2.9B | 6 | 768 | 6 | DCLM+code+math, C4 |
| 0.16B | 3.13e+18 | 3.2B | 6 | 768 | 6 | DCLM+code+math, C4 |
| 0.16B | 3.44e+18 | 3.6B | 6 | 768 | 6 | DCLM+code+math, C4 |
| 0.16B | 1.51e+18 | 1.6B | 6 | 768 | 6 | DCLM+code+math |
| 0.16B | 6.49e+18 | 6.7B | 6 | 768 | 6 | DCLM+code+math |
| 0.12B | 7.02e+18 | 9.5B | 5 | 640 | 5 | DCLM+code+math |
| 0.12B | 1.63e+18 | 2.2B | 5 | 640 | 5 | DCLM+code+math, C4 |
| 0.12B | 1.79e+18 | 2.4B | 5 | 640 | 5 | DCLM+code+math, C4 |
| 0.12B | 1.97e+18 | 2.7B | 5 | 640 | 5 | DCLM+code+math, C4 |
| 0.12B | 9.51e+17 | 1.3B | 5 | 640 | 5 | DCLM+code+math |
| 0.12B | 4.10e+18 | 5.6B | 5 | 640 | 5 | DCLM+code+math |
| 0.12B | 3.38e+18 | 4.6B | 5 | 640 | 5 | DCLM+code+math |
| 0.12B | 1.46e+19 | 19.8B | 5 | 640 | 5 | DCLM+code+math |
| 0.09B | 8.94e+18 | 16.3B | 4 | 512 | 4 | DCLM+code+math |
| 0.09B | 1.10e+18 | 2.0B | 4 | 512 | 4 | DCLM+code+math, C4 |
| 0.09B | 7.38e+18 | 13.5B | 4 | 512 | 4 | DCLM+code+math |
| 0.09B | 3.56e+18 | 6.5B | 4 | 512 | 4 | DCLM+code+math |
| 0.09B | 1.00e+18 | 1.8B | 4 | 512 | 4 | DCLM+code+math, C4 |
| 0.09B | 2.08e+18 | 3.8B | 4 | 512 | 4 | DCLM+code+math |
| 0.09B | 4.31e+18 | 7.9B | 4 | 512 | 4 | DCLM+code+math |
| 0.09B | 1.71e+18 | 3.1B | 4 | 512 | 4 | DCLM+code+math |
| 0.06B | 9.96e+17 | 2.6B | 3 | 384 | 3 | DCLM+code+math |
| 0.06B | 4.29e+18 | 11.1B | 3 | 384 | 3 | DCLM+code+math |
| 0.06B | 2.07e+18 | 5.4B | 3 | 384 | 3 | DCLM+code+math |
| 0.06B | 3.54e+18 | 9.2B | 3 | 384 | 3 | DCLM+code+math |
| 0.06B | 1.71e+18 | 4.4B | 3 | 384 | 3 | DCLM+code+math |
| 0.04B | 1.27e+18 | 5.2B | 2 | 256 | 2 | DCLM+code+math |
| 0.04B | 8.96e+17 | 3.7B | 2 | 256 | 2 | DCLM+code+math |
| 0.04B | 1.53e+18 | 6.3B | 2 | 256 | 2 | DCLM+code+math |
| 0.04B | 1.86e+18 | 7.6B | 2 | 256 | 2 | DCLM+code+math |

## C   Upper and Lower Bounds for Pass@k

We derive analytical bounds for the pass@k probability, i.e. the probability that at least one of $k$ independently sampled attempts succeeds. Let each trial succeed with probability $q \in [0, 1]$ independently. Then the probability that all $k$ trials fail is

$$p(\text{All Failures up to } k) = (1 - q)^k,$$

and consequently, the probability of at least one success (the *pass@k*) is

$$p(\text{pass@}k) = 1 - (1 - q)^k. \tag{6}$$

For $x > -1$ and integer $k \geq 1$, Bernoulli's inequality states that

$$(1 + x)^k \geq 1 + kx.$$

Setting $x = -q$ (so that $1 + x = 1 - q$), we obtain

$$(1 - q)^k \geq 1 - kq.$$

Substituting this into Equation 6 gives the *upper bound*:

$$p(\text{pass@}k) \leq \max(kq, 1). \tag{7}$$

Using the classical inequality $1 - q \leq e^{-q}$ for $q \geq 0$, we have

$$(1 - q)^k \leq e^{-kq}.$$

Substituting this into Equation 6, and noting that for any $x > 0$ with Taylor expansion we have $e^x \geq 1 = x$, thus $xe^{-x} \leq 1 - e^{-x}$, yields the *lower bound* by setting $x = kq$:

$$p(\text{pass@}k) \geq 1 - e^{-kq} \geq kqe^{-kq}. \tag{8}$$

Combining Equation 7 and Equation 8, we obtain

$$kqe^{-kq} \leq p(\text{pass@}k) \leq \max(kq, 1). \tag{9}$$

For a model with compute budget $C$, we can now use our approximation of $q \approx e^{-\frac{A}{C^\alpha}}$. The lower bound here motivates our formula of Equation 5. We have

$$p(\text{pass@}k) = Q(C, k) \approx kqe^{-kq} = k \exp\left(-k \exp\left(-\frac{A}{C^\alpha}\right) - \frac{A}{C^\alpha}\right),$$

$$-\log Q(C, K) \approx \log k + \frac{A}{C^\alpha} + k \exp(-\frac{A}{C^\alpha}),$$

$$log\left(-\log Q(C, K)\right) \approx \log(A) + \alpha \log(C) + \frac{A}{C^\alpha} \log k + \log \log k$$

where in Equation 5, we ignore the effect of $\log \log k$ and we modify the interaction term of $k$ and $C$ to a simpler form of $\log C \log k$. We additionally learn each of the coefficient of the equation rather than assuming they all follow the same $A$ and $\alpha$.

## D    More Details for the Choice of Proxy Metric in Two-Stage Approach

### D.1    Most Proxy Metrics Show Strong Predictive Power

We evaluate the efficacy of various proxy metrics in predicting final downstream performance on several benchmarks. Following the methodology in Section 4.2.1, we model the relationship between each proxy and downstream accuracy using a logistic function. As shown in Table 9, the low RMSE and high $R^2$ values indicate that nearly all proxy metrics are strong predictors of the final outcome.

This key result implies that the specific choice of proxy metric is less critical than previously assumed, affording practitioners the flexibility to choose based on convenience or computational cost. More importantly, it suggests that if many proxies are equally predictive, a direct optimization of the downstream task might be a more effective approach. We validate this hypothesis experimentally: our direct approach not only performs well but consistently outperforms all proxy-based methods.

Additionally, we analyze the predictive power of individual MMLU sub-tasks for overall MMLU accuracy (Table 10). We find considerable variance, indicating that certain sub-tasks are far better predictors of aggregate performance than others.

### D.2    Comparing Scaling Laws across Benchmarks

We evaluate the performance of two-stage (linear, logistic) and direct (BNSL, simple power law) modeling approaches on several benchmarks. As illustrated in Figure 7, while all methods demonstrate a high goodness-of-fit, their performance differs markedly during extrapolation.

The direct approaches consistently outperform their two-stage counterparts. We attribute this performance gap to the compounding errors inherent in two-stage pipelines, where inaccuracies from the initial fitting stage are propagated and amplified. This finding is robust across all tested benchmarks, highlighting the benefits of a direct, end-to-end modeling strategy for this task.

Table 9: Predictive performance of various metrics across all benchmarks. For each benchmark, we report the coefficient of determination ($R^2$), Root Mean Square Error (RMSE), Mean Absolute Error (MAE), and Mean Relative Error (MRE) when predicting the final task accuracy.

| Benchmark | Predictor Metric | $R^2$ | RMSE | MAE | MRE |
|---|---|---|---|---|---|
| arc_easy | arc_easy_acc | 0.996 | 0.008 | 0.006 | 0.013 |
| | arc_easy_brier | 0.997 | 0.008 | 0.006 | 0.012 |
| | arc_easy_loglikelihood | 0.995 | 0.010 | 0.007 | 0.014 |
| | arc_easy_neg_loglikelihood_norm | 0.989 | 0.013 | 0.010 | 0.021 |
| | c4_loss | 0.993 | 0.011 | 0.009 | 0.018 |
| | dclm_loss | 0.993 | 0.011 | 0.009 | 0.018 |
| | fineweb_loss | 0.993 | 0.011 | 0.009 | 0.018 |
| | mmlu_5s_loss | 0.992 | 0.012 | 0.009 | 0.018 |
| | openwebtext_2_loss | 0.992 | 0.012 | 0.009 | 0.019 |
| arc_challenge | arc_challenge_acc | 0.993 | 0.008 | 0.006 | 0.021 |
| | arc_challenge_brier | 0.994 | 0.007 | 0.006 | 0.020 |
| | arc_challenge_loglikelihood | 0.987 | 0.011 | 0.008 | 0.028 |
| | arc_challenge_neg_loglikelihood_norm | 0.980 | 0.012 | 0.009 | 0.032 |
| | c4_loss | 0.991 | 0.009 | 0.007 | 0.025 |
| | dclm_loss | 0.991 | 0.009 | 0.007 | 0.025 |
| | fineweb_loss | 0.991 | 0.009 | 0.007 | 0.024 |
| | mmlu_5s_loss | 0.990 | 0.009 | 0.007 | 0.025 |
| | openwebtext_2_loss | 0.989 | 0.010 | 0.008 | 0.027 |
| sciq | c4_loss | 0.985 | 0.015 | 0.012 | 0.017 |
| | dclm_loss | 0.985 | 0.016 | 0.012 | 0.017 |
| | fineweb_loss | 0.985 | 0.015 | 0.012 | 0.017 |
| | mmlu_5s_loss | 0.987 | 0.015 | 0.012 | 0.016 |
| | openwebtext_2_loss | 0.985 | 0.016 | 0.012 | 0.016 |
| | sciq_acc | 0.992 | 0.011 | 0.009 | 0.013 |
| | sciq_brier | 0.994 | 0.010 | 0.008 | 0.011 |
| | sciq_loglikelihood | 0.984 | 0.016 | 0.013 | 0.018 |
| | sciq_neg_loglikelihood_norm | 0.978 | 0.018 | 0.014 | 0.021 |
| piqa | c4_loss | 0.994 | 0.005 | 0.004 | 0.006 |
| | dclm_loss | 0.994 | 0.005 | 0.004 | 0.006 |
| | fineweb_loss | 0.994 | 0.005 | 0.004 | 0.006 |
| | mmlu_5s_loss | 0.994 | 0.006 | 0.005 | 0.007 |
| | openwebtext_2_loss | 0.994 | 0.006 | 0.004 | 0.007 |
| | piqa_acc | 0.992 | 0.007 | 0.005 | 0.008 |
| | piqa_brier | 0.995 | 0.005 | 0.004 | 0.006 |
| | piqa_loglikelihood | 0.993 | 0.006 | 0.005 | 0.007 |
| | piqa_neg_loglikelihood_norm | 0.986 | 0.008 | 0.006 | 0.010 |
| hellaswag | c4_loss | 0.974 | 0.016 | 0.012 | 0.039 |
| | dclm_loss | 0.975 | 0.015 | 0.011 | 0.039 |
| | fineweb_loss | 0.974 | 0.016 | 0.012 | 0.039 |
| | mmlu_5s_loss | 0.969 | 0.017 | 0.012 | 0.041 |
| | openwebtext_2_loss | 0.977 | 0.015 | 0.011 | 0.037 |
| hellaswag | c4_loss | 1.000 | 0.002 | 0.001 | 0.004 |
| | dclm_loss | 1.000 | 0.002 | 0.002 | 0.004 |
| | fineweb_loss | 1.000 | 0.002 | 0.001 | 0.004 |
| | hellaswag_brier | 0.000 | 0.108 | 0.096 | 0.260 |
| | hellaswag_loglikelihood | 1.000 | 0.002 | 0.002 | 0.004 |
| | hellaswag_neg_loglikelihood_norm | 1.000 | 0.002 | 0.002 | 0.004 |
| | mmlu_5s_loss | 0.999 | 0.004 | 0.003 | 0.007 |
| | openwebtext_2_loss | 0.998 | 0.004 | 0.003 | 0.009 |

Table 10: Predictive performance of in-domain sub-task accuracy for MMLU.

| Predictor Metric | $R^2$ | RMSE | MAE | MRE |
|---|---|---|---|---|
| Abstract Algebra | 0.521 | 0.068 | 0.042 | 0.125 |
| Anatomy | 0.932 | 0.025 | 0.017 | 0.052 |
| Astronomy | 0.969 | 0.017 | 0.013 | 0.043 |
| Business Ethics | 0.938 | 0.024 | 0.016 | 0.050 |
| Clinical Knowledge | 0.972 | 0.016 | 0.012 | 0.040 |
| College Biology | 0.950 | 0.022 | 0.016 | 0.051 |
| College Chemistry | 0.400 | 0.076 | 0.052 | 0.158 |
| College Computer Science | 0.838 | 0.039 | 0.024 | 0.073 |
| College Mathematics | 0.759 | 0.048 | 0.029 | 0.088 |
| College Medicine | 0.909 | 0.029 | 0.020 | 0.062 |
| College Physics | 0.442 | 0.073 | 0.050 | 0.152 |
| Computer Security | 0.972 | 0.016 | 0.012 | 0.039 |
| Conceptual Physics | 0.921 | 0.027 | 0.019 | 0.062 |
| Econometrics | 0.525 | 0.067 | 0.043 | 0.131 |
| Electrical Engineering | 0.940 | 0.024 | 0.014 | 0.042 |
| Elementary Mathematics | 0.899 | 0.031 | 0.021 | 0.065 |
| Formal Logic | 0.495 | 0.069 | 0.046 | 0.142 |
| Global Facts | 0.240 | 0.085 | 0.059 | 0.179 |
| High School Biology | 0.978 | 0.014 | 0.011 | 0.038 |
| High School Chemistry | 0.874 | 0.035 | 0.023 | 0.073 |
| High School Computer Science | 0.892 | 0.032 | 0.022 | 0.069 |
| High School European History | 0.974 | 0.016 | 0.011 | 0.037 |
| High School Geography | 0.978 | 0.014 | 0.011 | 0.036 |
| High School Government And Politics | 0.982 | 0.013 | 0.010 | 0.033 |
| High School Macroeconomics | 0.957 | 0.020 | 0.014 | 0.047 |
| High School Mathematics | 0.000 | 0.098 | 0.070 | 0.209 |
| High School Microeconomics | 0.948 | 0.022 | 0.017 | 0.056 |
| High School Physics | 0.704 | 0.053 | 0.034 | 0.107 |
| High School Psychology | 0.986 | 0.012 | 0.009 | 0.029 |
| High School Statistics | 0.704 | 0.053 | 0.034 | 0.105 |
| High School Us History | 0.963 | 0.019 | 0.015 | 0.052 |
| High School World History | 0.969 | 0.017 | 0.014 | 0.048 |
| Human Aging | 0.933 | 0.025 | 0.018 | 0.060 |
| Human Sexuality | 0.970 | 0.017 | 0.012 | 0.036 |
| International Law | 0.952 | 0.021 | 0.015 | 0.048 |
| Jurisprudence | 0.958 | 0.020 | 0.015 | 0.050 |
| Logical Fallacies | 0.951 | 0.022 | 0.017 | 0.057 |
| Machine Learning | 0.573 | 0.064 | 0.040 | 0.123 |
| Management | 0.947 | 0.023 | 0.017 | 0.055 |
| Marketing | 0.970 | 0.017 | 0.013 | 0.047 |
| Medical Genetics | 0.917 | 0.028 | 0.020 | 0.065 |
| Miscellaneous | 0.984 | 0.012 | 0.010 | 0.037 |
| Moral Disputes | 0.979 | 0.014 | 0.011 | 0.038 |
| Moral Scenarios | 0.000 | 0.098 | 0.070 | 0.209 |
| Nutrition | 0.982 | 0.013 | 0.009 | 0.030 |
| Philosophy | 0.969 | 0.017 | 0.013 | 0.043 |
| Prehistory | 0.979 | 0.014 | 0.011 | 0.037 |
| Professional Accounting | 0.914 | 0.029 | 0.020 | 0.064 |
| Professional Law | 0.000 | 0.098 | 0.070 | 0.209 |
| Professional Medicine | 0.873 | 0.035 | 0.021 | 0.066 |
| Professional Psychology | 0.970 | 0.017 | 0.014 | 0.048 |
| Public Relations | 0.946 | 0.023 | 0.016 | 0.053 |
| Security Studies | 0.967 | 0.018 | 0.012 | 0.039 |
| Sociology | 0.987 | 0.011 | 0.009 | 0.029 |
| Us Foreign Policy | 0.974 | 0.016 | 0.012 | 0.040 |
| Virology | 0.947 | 0.023 | 0.015 | 0.048 |
| World Religions | 0.957 | 0.020 | 0.015 | 0.052 |

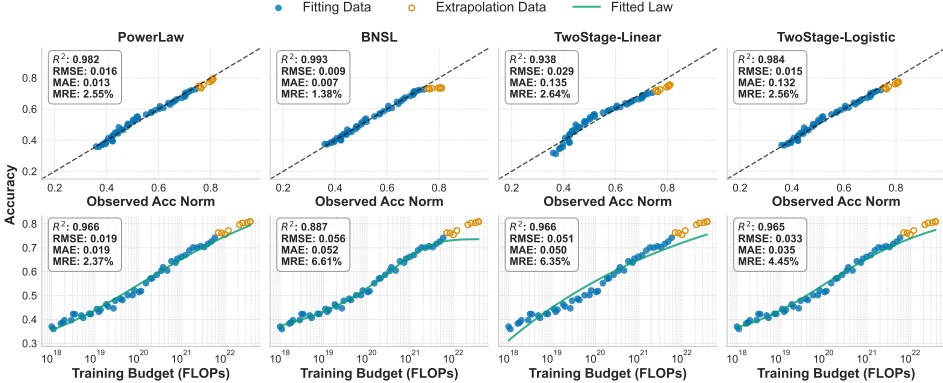

(a) ARC Easy. Legend depicts error on fitted points (top row) and validation points (bottom row).

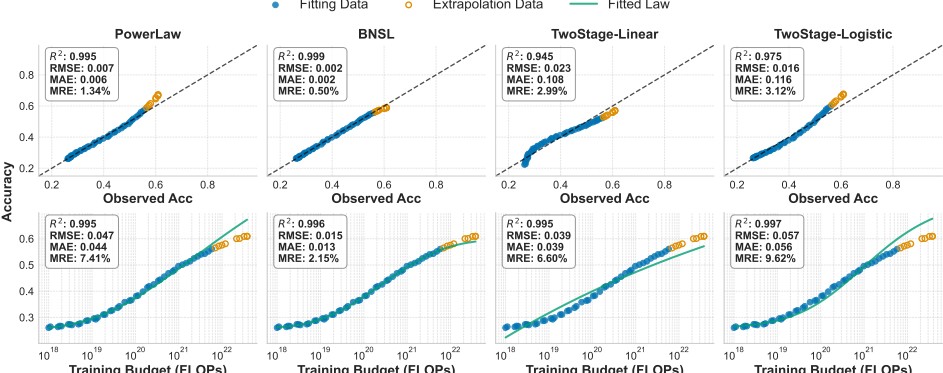

(b) HellaSwag. Legend depicts error on fitted points (top row) and validation points (bottom row).

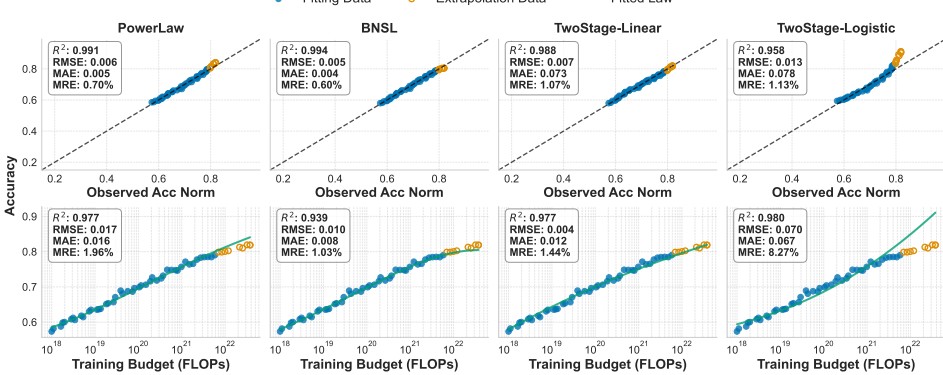

(c) PIQA. Legend depicts error on fitted points (top row) and validation points (bottom row).

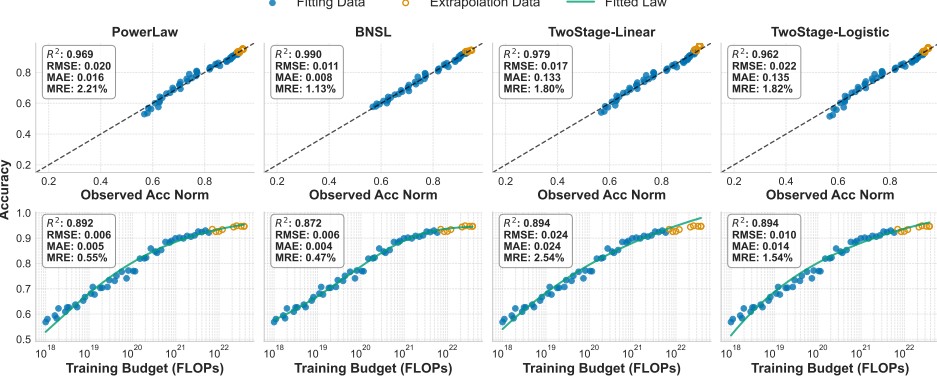

(d) SciQ. Legend depicts error on fitted points (top row) and validation points (bottom row).

Figure 7: Comparing the direct approaches (BNSL, simple power law) with two-stage approaches (Linear and Logistic) for various benchmarks.

# E CRITICAL FLOPS THRESHOLD FOR PREDICTING DOWNSTREAM PERFORMANCE

The choice of the FLOPs threshold, which partitions data for training and validation, significantly impacts the reported performance and reliability of scaling law models. We conduct an analysis to quantify this sensitivity and determine the minimum FLOPs threshold required for each model to achieve a robust extrapolation, which we define as achieving a Mean Relative Error (MRE) below 10%. In Figure 8, we plot the extrapolation MRE and MAE as a function of the FLOPs threshold. The results show that the BNSL and simple power law models exhibit stable performance, maintaining low MRE across a wide range of thresholds. In contrast, the two-stage linear and logistic models are less robust, demonstrating higher error rates and greater sensitivity to the choice of threshold.

To formalize this robustness analysis, we introduce a binary success criterion: a model "succeeds" at a given threshold if its MRE is below 10%. As illustrated in Figure 9, we sweep the FLOPs threshold from $6e19$ to $5e22$ and fit a logistic regression to the binary success outcomes. This allows us to estimate the FLOPs threshold at which each model is likely to fail. Our findings confirm that the simple power law is the most resilient model, followed closely by BNSL. The two-stage models are shown to be suboptimal, requiring a much higher FLOPs threshold to consistently achieve reliable fits. This analysis provides a way to assess model stability for scaling law extrapolation.

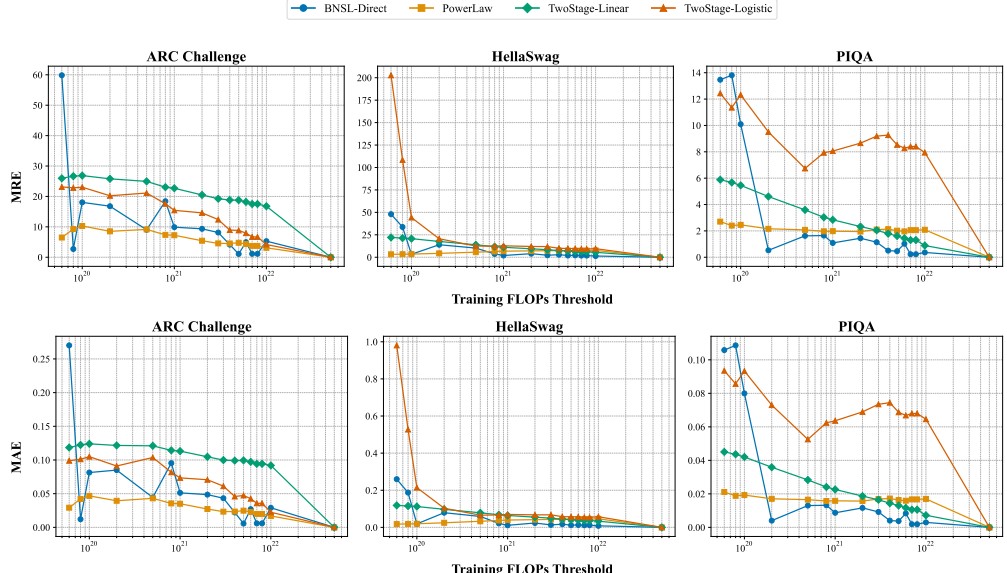

Figure 8: Trend of two metrics MRE and MAE across benchmarks and scaling laws.

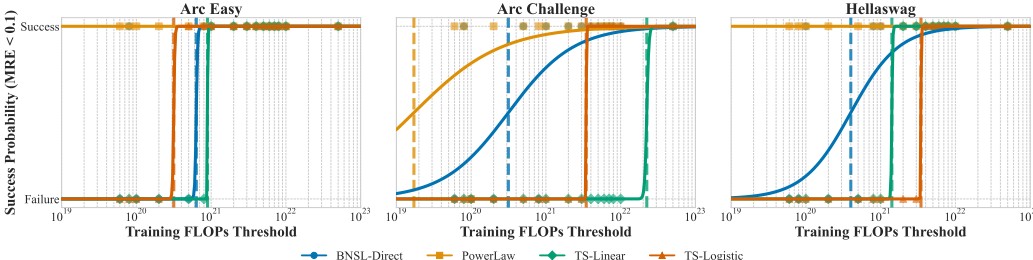

Figure 9: Minimum FLOPs Threshold for Reliable Extrapolation. We define 'Success' at a given FLOPs threshold as achieving MRE below 10%. A logistic regression is fitted to these binary outcomes to model the probability of success as a function of the threshold. The vertical dashed line indicates the estimated threshold where the probability of success surpasses 50%, effectively separating the failure and success regimes.

## F  DOWNSTREAM SCALING CURVES FOR C4 DATASET

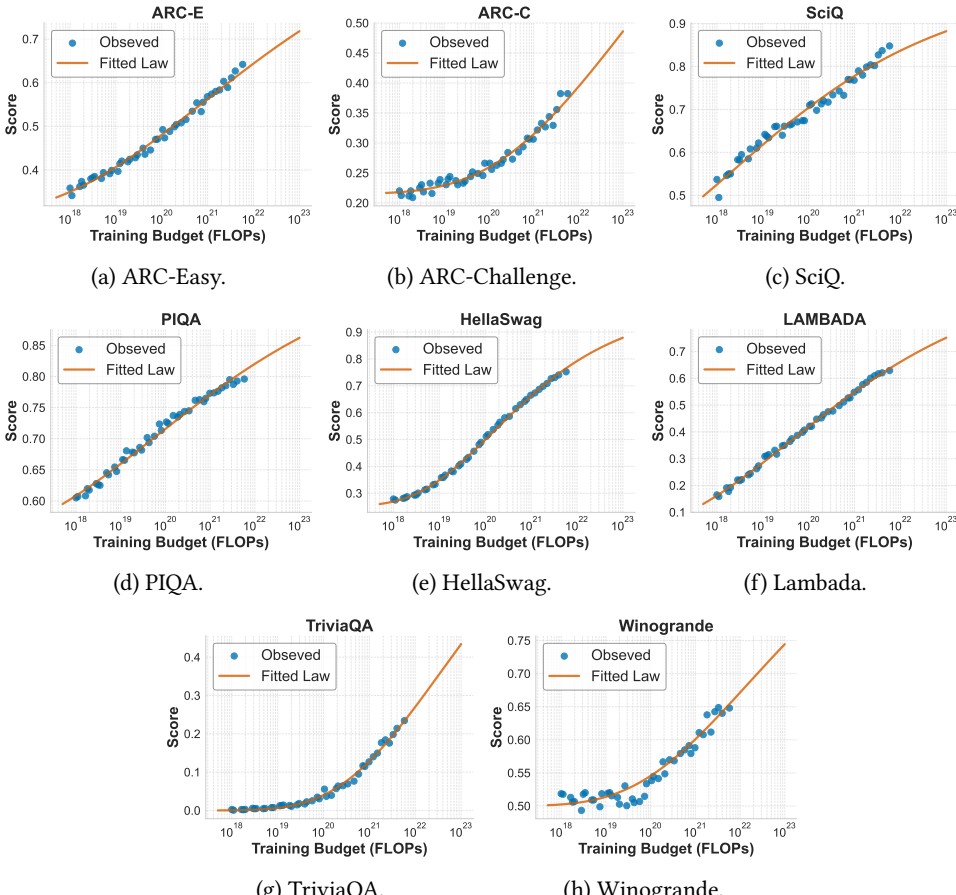

Figure 10: Downstream scaling curves when changing the pre-training dataset to C4.

## G  HYPERPARAMETERS

We determined the maximum batch size with near-optimal performance at 1.8B training tokens as 64 and scaled it proportionally to $D^{0.5}$, based on the recommendations from the literature (Filatov et al., 2025; Bergsma et al., 2025; Zhang et al., 2025). We set the maximum global learning rate to 5e-3, derived by tuning a proxy model and later transferring with $\mu$-parametrization (Yang et al., 2022) in its simplified form, as described in Wortsman et al. (2023). We follow Gunter et al. (2024) in the setup of optimizer and weight decay (decoupled weight decay of 3.16e-4), since it has been ablated for quality and stability across a range of compute scales.

## H  PREDICTING THE AVERAGE ACROSS BENCHMARKS

Gadre et al. (2024b) propose to predict the average score across tasks rather than metrics on specific benchmarks. They use the two stage approach in this problem. Here we present the initial examination on the possibility of describing and extrapolating the value the average score using the direct method.

We note that is not immediately clear which functional form is theoretically correct, as we take the mean of the scores on multiple individual benchmarks. Taking the empirical perspective, in this pilot study we examine whether the functional form of Equation 2 can be used in this case.

We consider models trained with the to token-to-param ratio 20. For each experiment, we calculate the mean of the model scores across Arc-E, Arc,C, SciQ, PIQA, HellaSwag, WinoGrande, WebQS, TriviaQA, Lambada and HumanEval. We do not consider LBPP, as this metric only gives signal on a relatively large scale, and would not contribute meaningfully. We exclude models where any of the benchmarks achieved a score of less than 5% points above the random performance to reduce the points with random noise. For all metrics, where the scores are not by default in the range [0, 1] (i.e. multiple-choice tasks), before taking the mean, we first normalize the scores as described in Section 3.2 (we apply the transformation $Q' := (Q - Q_{\mathrm{random}})/(1 - Q_{\mathrm{random}})$).

The results are illustrated in Figure 11. We exclude expeirments with more than 6e21 training FLOPs as the set for validating the extrapolation. We observe good quality of the fit, as outlined in Table 11. As the next step, we could consider modeling this relationship using an equation with more fitted parameters, like BNSL with multiple breaking points. We leave further examination of this setup for future work.

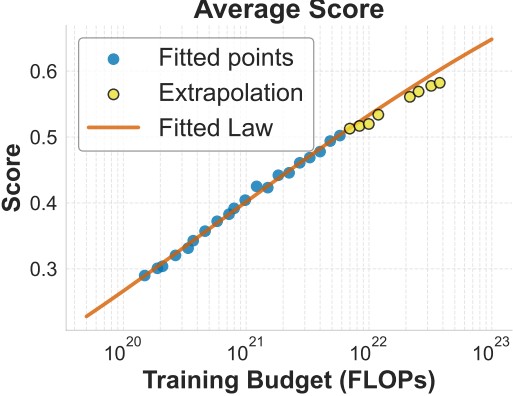

Figure 11: Illustration of the fit quality for predicting the average across benchmarks.

Table 11: Fit quality of predicting the average across benchmarks.

| Train MAE | Valid MAE | Train MRE (%) | Valid MRE (%) |
|-----------|-----------|---------------|---------------|
| 0.0030    | 0.0116    | 0.76%         | 2.08%         |

## I    Modeling Accuracy with Irreducible Error

In Section 3 we assume the perfect scenario, where the maximum achievable accuracy on each task is equal to 1. This assumption may not hold in practice: it has been observed that the benchmark score plateaus on a value below 1, due to a certain number of incorrectly labeled or ambiguous questions (Vendrow et al., 2025). Here we examine the possibility of adding the asymptote of maximum achievable accuracy to the functional form.

We replicate the approach secribed in Section 3.2, but with Equation 2 modified to incorporate the irreducible error:

$$-\log(Q) = \frac{A}{C^\alpha} + E, \qquad (10)$$

where $Q_{\mathrm{max}} := \exp(-E) \in (0, 1)$ represents the estimated value of maximum achievable accuracy. We fit the coefficients $A$, $\alpha$ and $E$, using only models with training FLOPs less than 6e21, leaving the remaining ones as the validation set. The results are illustrated in Figure 12, showing a good quality of the fit with this functional form.

Table 12 outlines the estimated values of $Q_{\mathrm{max}}$ and error rates on the validation points. For seven benchmarks, the fitted value of $Q_{\mathrm{max}}$ remains unchanged from Equation 2, with $Q_{\mathrm{max}} = 1$. In three cases (PIQA, HellaSwag, Lambada), we can see meaningful values of maximum accuracy, with $Q_{\mathrm{max}}$

estimated as 0.903, 0.912 and 0.947, respectively. For two benchmarks, where the accuracy observed in all experiments used for the fit is relatively far from perfect score (Winogrande, WebQS), the estimated values of $Q_{\max}$ (0.776, 0.51) are unlikely to be correct.

To summarize our findings, using the functional form with the $Q_{\max}$ can be a useful strategy, especially when the target models are expected to approach the maximum achievable accuracy. However, this approach must be applied with caution, to make sure that the estimated perfect score aligns with the number of incorrect or ambiguous questions. In all cases, it is crucial to carefully examine the evaluation datasets, and apply both manual and automatic quality filtering (for example, as descibed in Vendrow et al. (2025)), to ensure that the test scores reliably measure model quality.

We can incorporate the maximum achievable score in other scaling law forms considered in this work, similarly to how we did it for Equation 2. For example, in Equation 4, we can analogously incorporate the additional summand, resulting in the following functional form: $-\log Q = \frac{A}{N^\alpha} + \frac{B}{D^\beta} + E$.

Table 12: Estimated maximum accuracy and error rates for fitting the scaling law with maximum achievable accuracy.

| Benchmark | Metric Type | Fitted $Q_{\max}$ | Valid MRE | Valid MAE |
|---|---|---|---|---|
| ARC-E | acc norm | 1.000 | 1.81% | 0.0143 |
| ARC-C | acc norm | 1.000 | 1.26% | 0.0066 |
| SciQ | acc norm | 1.000 | 0.56% | 0.0053 |
| PIQA | acc norm | 0.903 | 0.30% | 0.0024 |
| HellaSwag | acc norm | 0.912 | 0.99% | 0.0079 |
| Winogrande | acc | 0.776 | 1.65% | 0.0122 |
| WebQS | exact match | 0.510 | 1.26% | 0.0028 |
| TriviaQA | exact match | 1.000 | 6.93% | 0.0290 |
| LAMBADA | acc | 0.947 | 2.27% | 0.0170 |
| GSM8K | exact match | 1.000 | 10.99% | 0.0635 |
| HumanEval | pass@1 | 1.000 | 3.82% | 0.0125 |
| LBPP | pass@1 | 1.000 | 6.17% | 0.0070 |

## J  ADDITIONAL DETAILS ON SCALING LAW FITTING AND VALIDATION

We fit Equation 1 by optimizing the Huber Loss ($\delta = 1e - 3$) using the basin-hopping (Wales & Doye, 1997) algorithm. We use only points with accuracy $Q > Q_{\text{random}}$ for the fit of BNSL (note that we apply a less strict filtering than $Q_{\text{random}} + 0.05$ considered for Equation 2, since this helps in estimating the lower asymptote and improves extrapolation performance for BNSL). In the case of LBPP, we filter the fitted points to $Q_{\text{random}} + 0.02$, since all of the models at our scale achieve relatively low scores on this benchmark and we want to ensure there are enough points to consider. In fitting Equation 10 and in Appendix H, we adopt the L-BFGS-B algorithm with the Huber Loss ($\delta = 1e - 3$) objective, sampling a grid of initalizations and choosing the one with the best score on the fitted points. Throughout the paper, we use least squares for fitting the coefficients in all other cases (Equation 2 and in TwoStage-Linear procedure), due to the closed form of the solution and lack of the dependency on hyperparameters in optimization. We detail the estimated values of $Q_{\text{random}}$ for the acc_norm in Table 13. For all other metric types, we adopt the standard random-guess baseline.

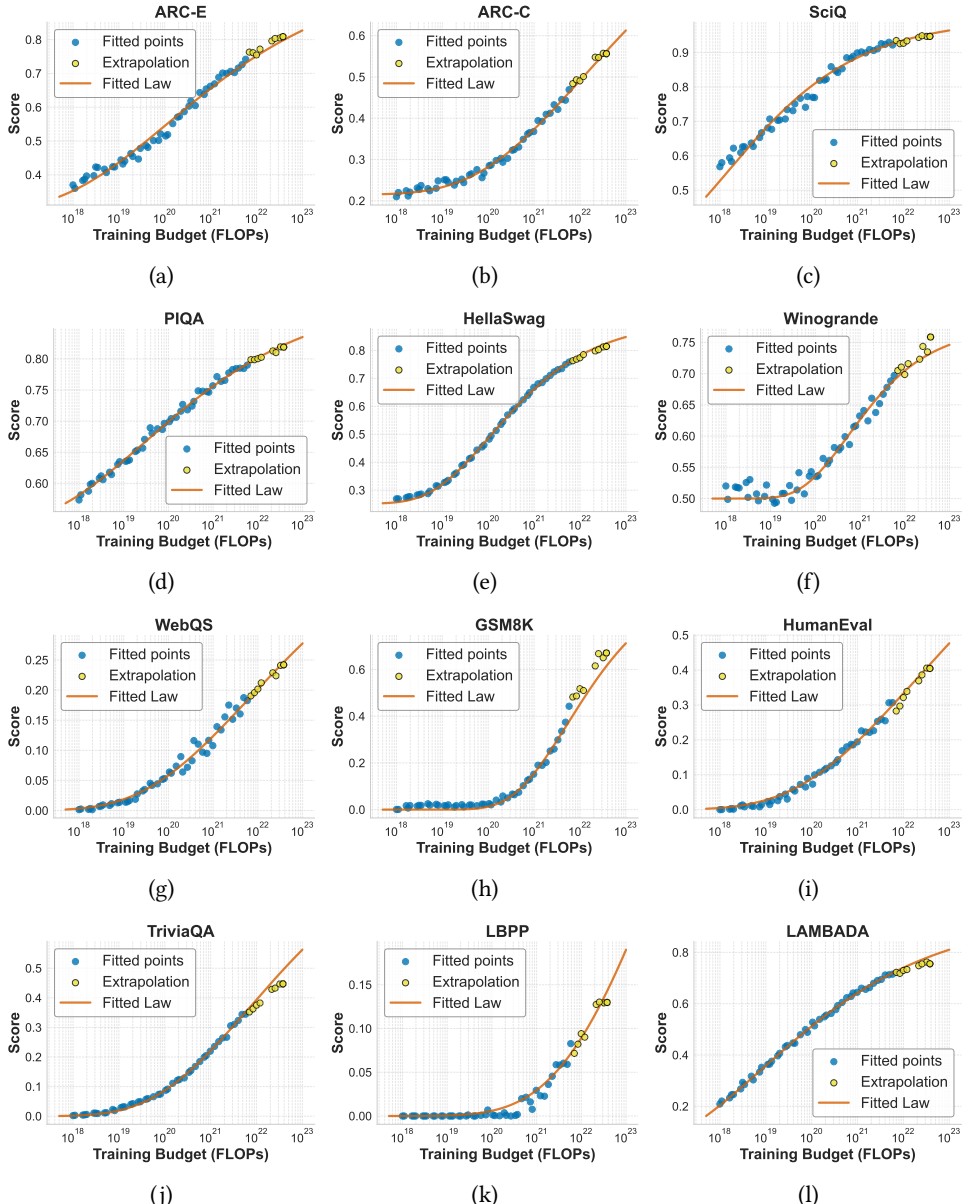

Figure 12: Results of the fitting of the equation with irreducible error for accuracy.

Table 13: Values of $Q_{\text{random}}$.

| Benchmark | Metric Type | $Q_{\text{random}}$ |
|---|---|---|
| Arc-E | acc_norm | 0.291 |
| Arc-C | acc_norm | 0.215 |
| SciQ | acc_norm | 0.304 |
| PIQA | acc_norm | 0.53 |
| HellaSwag | acc_norm | 0.252 |

# K   Fit Quality Plots for Varying Token to Param Ratio

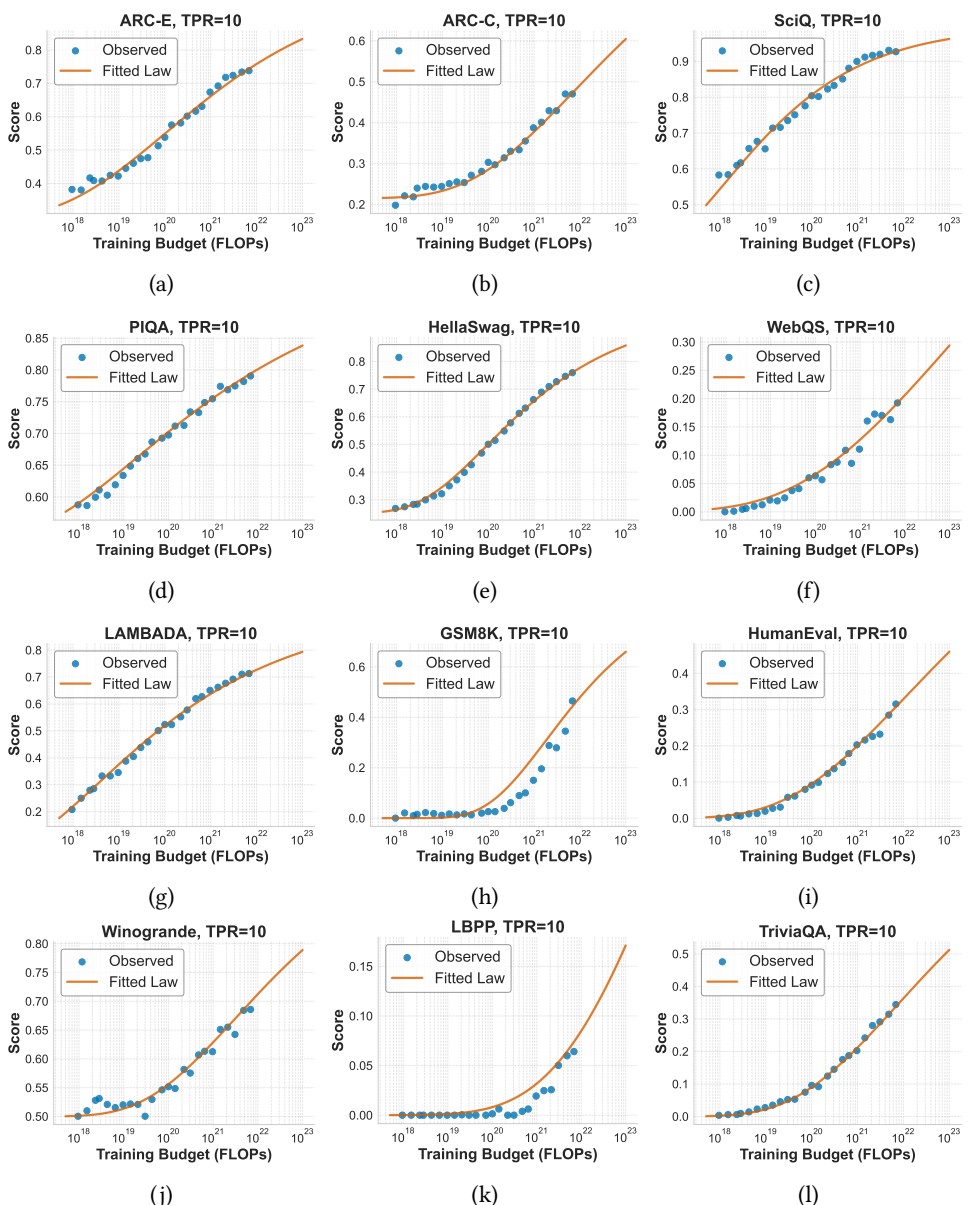

Figure 13: Fit of Eq. (4) for Token to Param Ratio 10.

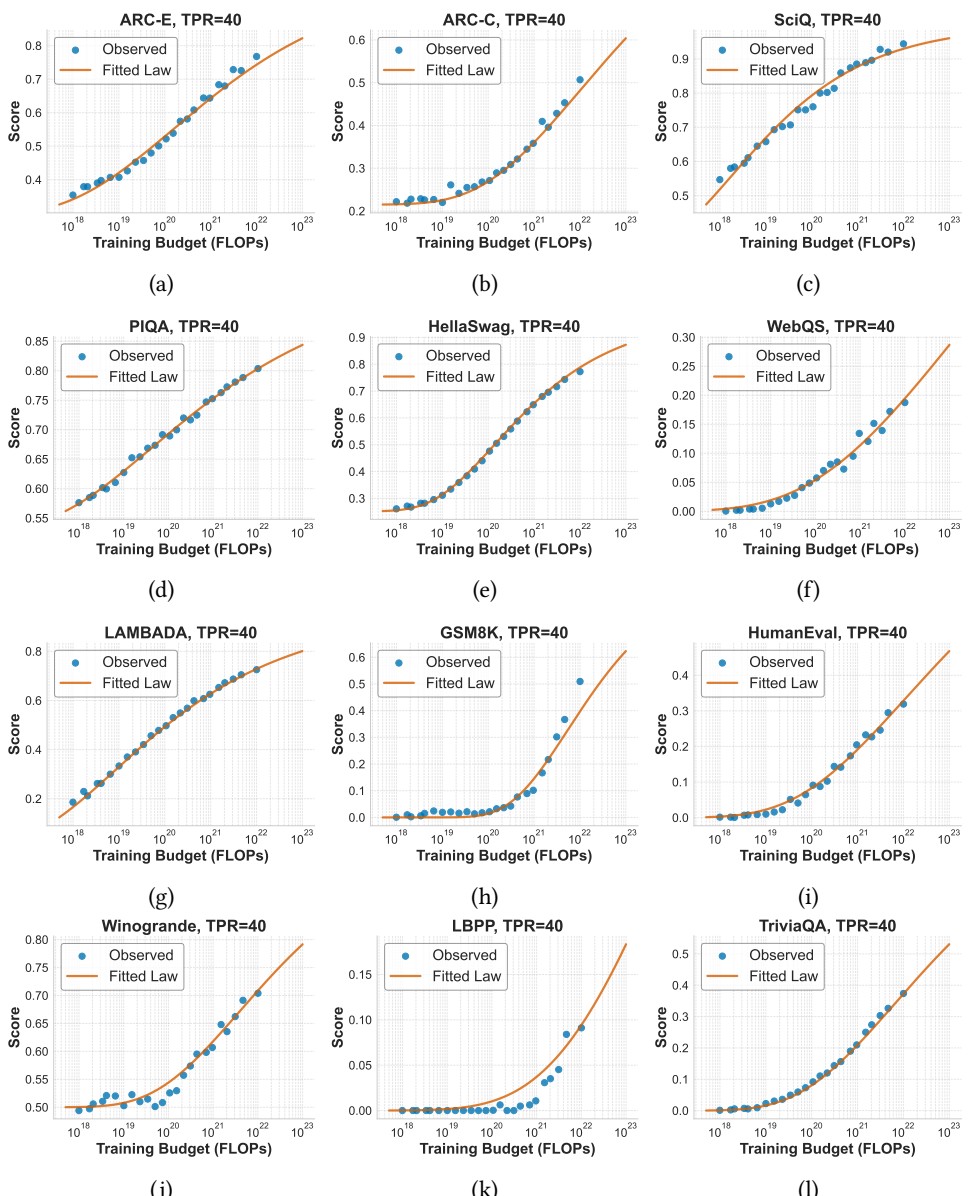

Figure 14: Fit of Eq. (4) for Token to Param Ratio 40.

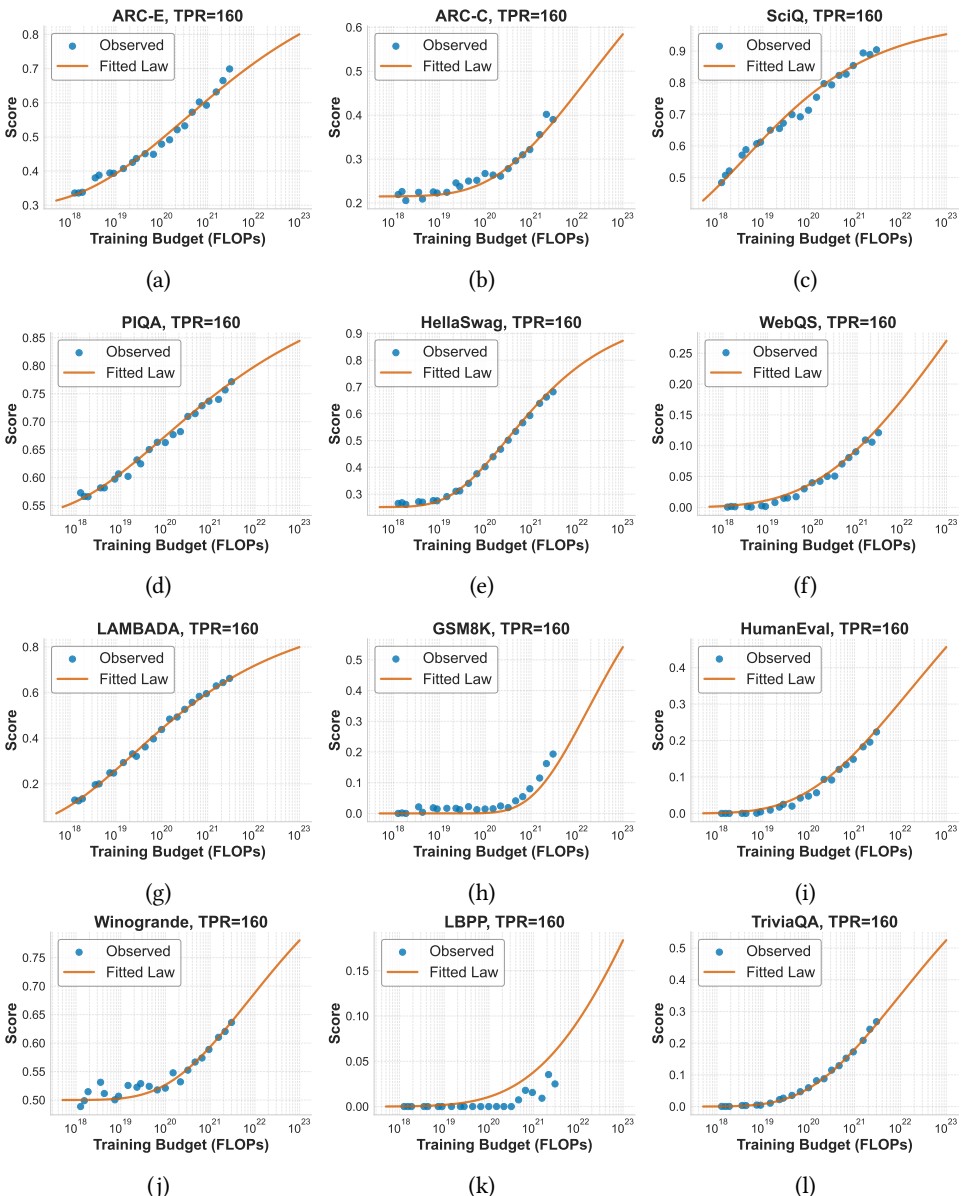

Figure 15: Fit of Eq. (4) for Token to Param Ratio 160.

