# OpenReview forum: "Revisiting the Scaling Properties of Downstream Metrics in Large Language Model Training"
_ICLR.cc/2026/Conference — ICLR 2026 Poster_

### Official Review · Reviewer_yiPV · 2025-10-28

**Soundness:** 3
**Presentation:** 3
**Contribution:** 3
**Rating:** 4
**Confidence:** 5

**Summary:**

The paper investigates the power of scaling laws in language models to directly predict downstream performance from models' parameter count and dataset size.  Using a functional form without irreducible error $-log(Q)=A/N^\alpha + B/D^\beta$ the authors fit scaling laws for 48 distinct training budgets and 5 token-to-parameter rations. Then they validate the predictive power of fitted scaling laws using error estimation on the held-out points. The results show that the direct approach of predicting downstream performance is a promising direction in exploring new ways to accurately and reliably predict language models' performance at large scale.

**Strengths:**

- One-stage approach to to directly predict downstream performance of LLMs from N and D.
- Two different approaches to downstream task modeling - BSNL and a simple power-law relationship.
- Authors validated their predictions using error on held-out points.
- Works for different downstream metrics: pass@k, accuracy etc.
- Baselines like two-stage approaches that are commonly used (FLOPs-to-NLL and then NLL-to-accuracy).
- Direct downstream prediction is more reliable and accurate than two-stage approaches.

**Weaknesses:**

- No code or data currently available which makes the reproduction and independent verification of the authors' claims impossible. The authors though promise to release the model losses and downstream evaluation results.
- Better performance of one-stage approach compared to two-stage one on individual benchmarks can be due to the fact that individual task performance may not be correlated to the specific downstream task performance (which also depends on the the validation set) and not only compounding errors from different fits (please see the questions section for more info).

**Questions:**

### Questions

- Are the checkpoints well tuned? The authors mentioned that they used optimizer setup from [1] as well as  $\mu P$ for hyperparameter transfer but they don't mention any hyperparameter tuning or at least their motivation to use specific hyperparameters.
- In [2] authors note that it's hard to predict single benchmark performance with their two-stage approach reliably while the average across 17 benchmarks can be predicted quite well. Using the one-stage approach described here, can the average of multiple downstream task accuracies be reliably predicted?
- In two-stage approaches (Fig. 7) what specific proxy metric was used?
- In table 9, we see high $R^2$ and RMSE for different proxies and downstream tasks, which should prove that all proxy metrics are strong predictors for downstream performance, while in the Fig. 7, fits with higher RMSE and $R^2$ do not necessarily have the best predictive power (e.g. in Fig. 7a BNSL has higher $R^2$ and RMSE than TwoStage-Logistic and yet TwoStage-Logistic has lower MAE and MRE on held-out points).  So does the statement *"most proxy metrics demonstrate strong predictive power for downstream task performance"* still hold?
- Why modelling downstream performance without irreducible error? In principle, yes it is possible to have 100% accuracy on a downstream task, however, in practice it is known that the real benchmark data can have ill-posed, mislabeled or ambiguous questions [3].


### References

1. Gunter, Tom, et al. "Apple intelligence foundation language models." _arXiv preprint arXiv:2407.21075_ (2024).
2. Gadre, Samir Yitzhak, et al. "Language models scale reliably with over-training and on downstream tasks." _arXiv preprint arXiv:2403.08540_ (2024).
3. Vendrow, Joshua, et al. "Do large language model benchmarks test reliability?." _arXiv preprint arXiv:2502.03461_ (2025).

---

> ### Author Response · Authors · 2025-11-17
> **Author Response (Part 1/2)**
>
> We thank the reviewer for the detailed and thorough feedback. We appreciate the recognition of the compatibility of the scaling law with multiple downstream metrics, validation of predictions on held-out points, and comparison with the baseline methods. Below we address the reviewer’s questions and mentioned weaknesses. If our response addresses the reviewer’s concerns, we kindly ask for the reconsideration of the paper score.
>
> We have updated the submitted article and supplementary materials to address the discussed questions. For the reviewer’s convenience, all changes in the updated paper are marked by using **green text font color**.
>
> **Code, data and reproducibility**. We agree that ensuring reproducibility is crucial for the research community. We would like to provide more details in that regard.
>
> - We train all models only on publicly available datasets and provide the details of the training data. We use either C4 (Section 5.1), or a more modern mixture consisting of 75% of tokens from DCLM, 15% from Stack v2, and 10% from OpenMathReasoning. We detail the dataset setup in the beginning of Section 3 (paragraph **Dataset**).
> - Our training code is based on an open-source repository, as detailed in Section 3 (paragraph **Code**).
> - Per the reviewer’s suggestion, we have updated our submission with a file containing final validation losses, proxy metric and downstream accuracy results of the considered models in the form of a csv file (submitted as supplementary material).
> - We also updated our submission with a zipped repository containing the code used for scaling law fitting.
> - Upon the acceptance of the paper, we will provide a link to the GitHub repository with scaling law fitting code and model evaluation results (available now as the supplementary material).
>
> We believe that the provided details will make the reproducibility of our results convenient and encourage further research in the community.
>
> **Hyperparameter tuning**. We thank the reviewer for the important question and apologize for the oversight in fully describing the hyperparameter tuning approach in the original submission. Ensuring that models are well tuned is a crucial step in deriving reliable scaling laws. In our experimental setup, we focus on two crucial hyperparameters: learning rate and batch size, while falling back to recipes validated in the literature in the remaining settings.
>
> We determined the maximum batch size with near-optimal performance at 1.8B training tokens as 0.26M tokens and scaled it proportionally to D^0.5, based on the recommendations from the literature [1] [2] [3]. We determined the optimal learning rate on the 1e17 training FLOPs scale as 5e-3, and transferred to larger models using $\mu$-parametrization. We follow [5] in all other hyperparameters, including weight decay, optimizer betas and epsilon (please refer to Appendix A in [5] for the detailed comparison with a more classical training setup).
>
> We placed the details of the hyperparameter setup in **Appendix G** of the updated submission.
>
> **Predicting the average score across benchmarks**. We examined the possibility of modeling the average benchmark score using our direct scaling law approach. We thank the reviewer for the suggestion to consider this setup and believe that including it will improve the completeness of our study. The results of this examination are provided in **Appendix H** of the updated submission. In summary, we observe a good quality of the fit when using the functional form of Equation 2 to describe the relationship between average score and training budget. We exclude all models with at least 6e21 training FLOPs as the validation set, observing valid MAE of 0.0116 and valid MRE of 2.08%, as outlined in Table 11 and visualized in Figure 11.

---

> ### Author Response · Authors · 2025-11-17
> **Author Response (Part 2/2)**
>
> **Irreducible error.** We would like to thank the reviewer for the valuable insight. In the original submission, we assumed a perfect scenario, where the maximum achievable score for each benchmark is equal to 1. However, as correctly pointed out by the reviewer, in practical cases many benchmarks have been shown to contain a certain number of incorrect or ambiguous questions, limiting the perfect score to a number below 1. To account for this effect, we extended our scaling law to consider the maximum achievable accuracy $Q_\text{max}$, with the details presented in **Appendix H** of the updated submission. We fit the coefficients in the equation and visualize results in Figure 12, while the estimated values of $Q_\text{max}$ are outlined in Table 12. Overall, accounting for $Q_\text{max}$ can be a useful tool improving the robustness of the scaling law when the scores are close to maximum accuracy. However, this approach must be used with caution, especially when all of the observed accuracy values are relatively far from the perfect score, and therefore estimating $Q_\text{max}$ is prone to errors. In all cases, we note the critical importance of careful design and filtering of the evaluation benchmarks to ensure their reliability.
>
> **Correction on the metrics in Fig. 7 and Table 9.** In Figure 7 (Appendix C) in the original submission, the top row contained RMSE and R2 **on the fitted points**, while the bottom row depicted MAE and MRE **on the extrapolation (validation) points**, resulting in discrepancy between the metrics. We realized that the legend of the figure was not properly described, causing confusion. We apologize for this oversight.
>
> To clarify, we updated the submission. Both Figure 7 and Table 9 now contain all four metrics: MAE, MRE, RMSE, and R2. Our observations do not change: we notice high correlation between each of the considered metrics with target accuracy, also as measured by MRE and MAE. Therefore, we do not expect the choice of the proxy metric to be the limiting factor in the case of the benchmarks considered in our work.
>
> However, even if there are cases where the intermediate step in the two stage approach is particularly problematic, this only underscores the benefits of our proposed direct approach. By modeling the benchmark accuracy directly, we remove both the dependence on the choice of the proxy metric and the effect of compounding errors.
>
> **Proxy metric used in two stage approach.** In Section 4.2, we use the negative normalized log-likelihood of the correct answer sequence (task loss) as the proxy metric, following [4]. However, as we conclude that other proxy metrics have a comparable predictive power, we expect similar results if other proxy metrics would be used.
>
>
> [1] Filatov et al., Time transfer: On optimal learning rate and batch size in the infinite data limit
>
> [2] Bergsma et al., Power lines: Scaling laws for weight decay and batch size in llm pre-training
>
> [3] Zhang et al., How does critical batch size scale in pre-training?
>
> [4] Bhagia et al., Establishing Task Scaling Laws via Compute-Efficient Model Ladders
>
> [5] Gunter et al., Apple Intelligence Foundation Language Models

---

> > ### Author Response · Authors · 2025-11-25
> >
> > Dear Reviewer yiPV,
> >
> > We appreciate the reviewer’s time and insightful comments. Based on the received feedback, we have updated the submission and responded in the rebuttal comment. Please let us know if there is anything we could add to improve the paper or clarify any concerns. Thank you!
> >
> > Authors

---

> > > ### Comment · Reviewer_yiPV · 2025-11-26
> > >
> > > Thank you for answering my questions and providing additional information. My concerns were addressed and I will revise my score.

---

### Official Review · Reviewer_MNqJ · 2025-11-01

**Soundness:** 3
**Presentation:** 3
**Contribution:** 2
**Rating:** 6
**Confidence:** 3

**Summary:**

This paper proposes to directly predict downstream task performance from training budget. The authors run experiments across a variety of benchmarks and compute budgets to show that a simple scaling law equation can model this relationship and outperforms prior work.

**Strengths:**

- wide coverage of benchmarks and types of tasks, various compute budgets (both scale and TPR)
- simplicity of approach, taking into account the nature of these benchmarks (e.g. S-shaped)
- results demonstrate some extrapolation to larger compute budgets
- compares directly with prior works (two stage approach)

**Weaknesses:**

- lack of motivation for practical usage of this compared to standard scaling laws
- scaling laws are often used to justify design decisions (e.g. architectural or dataset choices) - there is a lack of these alternatives and showing that the scaling laws preserve the ordering of the "better" design decision

**Questions:**

Irreducible term may be theoretically unnecessary, but benchmarks often have incorrect labels (e.g. MMLU). How would you account for this?

Could you better motivate why direct scaling laws to evaluation are practically more useful than perplexity alone?

---

> ### Author Response · Authors · 2025-11-17
>
> We thank the reviewer for the insightful comments and feedback. We appreciate the recognition of the wide coverage of benchmarks and tasks and direct comparison with prior works. Below we address the reviewer’s questions in detail. If the reviewer’s concerns are properly addressed, we would like to kindly ask for the reconsideration of the paper rating.
>
> To address the received feedback, we have updated our submission. For the reviewer's convenience, we marked all changes in the article using **green text font color**.
>
> **Irreducible term in modeling accuracy.** We thank the reviewer for the insightful comment. In the original submission, we assumed the ideal setting, where the accuracy on each benchmark approaches 1 if the training scale is large enough. However, we agree that in practice it has been observed that benchmarks often contain incorrect or ambiguous labels and questions. To account for this effect, in **Appendix H** of the updated submission we propose a version of the scaling law with the maximum achievable accuracy $Q_\text{max}$. We fit the coefficients in the equation and visualize results in Figure 12. The estimated values of $Q_\text{max}$ are outlined in Table 12. We conclude that this approach can be helpful in improving the robustness of extrapolation, especially when the observed or predicted scores approach the perfect accuracy. However, when all of the observed points include only models with relatively low scores, estimating the fraction of incorrect labels can be hard and prone to errors. Therefore, we emphasize the importance of careful inspection and filtering of the test examples to ensure their reliability in modeling the model quality, for example using the strategy described in [1].
>
> **Motivation and practical usage.** Standard scaling laws for training perplexity are established as the crucial tool in establishing large model training setups. However, there are questions which cannot be answered by modeling the loss alone. For example, when planning a large training run, we may want to know the minimum FLOPs budget needed to reach 50% on a given coding or math benchmark under a fixed architecture and dataset. Answering this kind of question requires describing and extrapolating the scaling of the metric of interest.
>
> The importance of this problem is also observed by researchers and engineers developing large models. For example, the GPT-4 report [2] notes: “Having a sense of the capabilities of a model before training can improve decisions around alignment, safety, and deployment.”. Similarly, authors of the Llama 3 paper [3] state: “In addition to determining the optimal model size, a major challenge is to forecast the flagship model’s performance on downstream benchmark tasks”. The dominant setup for solving this problem so far was to use the two stage procedure. In our paper, we show how to simplify and improve this approach with direct prediction. Therefore, we believe that our paper provides a useful tool for de-risking large training runs and constitutes a meaningful research contribution.
>
> We thank the reviewer for mentioning this important question. In the final version of our submission, we will extend the Introduction section to emphasize the motivation for our study.
>
> [1] Vendrow et al., Do large language model benchmarks test reliability?
>
> [2] OpenAI, GPT-4 Technical Report
>
> [3] Grattafiori et al., The Llama 3 Herd of Models

---

> > ### Author Response · Authors · 2025-11-25
> >
> > Dear Reviewer MNqJ,
> >
> > We are grateful for the insightful feedback and questions. We have updated our submission and responded in the rebuttal comment. Please let us know if there are any further changes we could make to improve the submission. Thank you!
> >
> > Authors

---

### Official Review · Reviewer_5erS · 2025-11-01

**Soundness:** 4
**Presentation:** 4
**Contribution:** 3
**Rating:** 8
**Confidence:** 3

**Summary:**

This paper proposes to directly model task accuracy as a function of compute budget, rather than relying on the usual two-stage approach where downstream performance is predicted through proxy metrics like pretraining loss, which often makes such predictions unreliable. The authors introduce a simple two-parameter scaling model under a fixed token-to-parameter ratio and show that it fits downstream performance well. They validate the framework on models up to 17B parameters trained on 350B tokens, and further demonstrate its ability to extrapolate and predict accuracy for models trained with up to 6.7× larger compute budgets.

**Strengths:**

The paper is clearly written, and I appreciate the measured and non-sensational tone throughout. The proposed accuracy scaling law fits the evaluated metrics well, showing strong consistency across scales. I also like that the paper carefully compares the one-stage approach to the traditional two-stage setup, showing lower MAE, MRE, and higher R² for the proposed method.

**Weaknesses:**

Please see Questions section.

**Questions:**

1. What do you think about applying the scaling law fits to predict a single aggregated metric, such as the average downstream accuracy across a broad set of tasks? Scaling laws are often used to understand and guide the training of generalist models that need to perform well across many tasks. Would your proposed fits extend to that setting, and how would you think about aggregating or weighting the different task metrics?
2. The scaling law fits for all benchmarks in the paper look very good, and I appreciate the mention of cases where non-monotonic behavior makes direct fitting difficult. I’m curious to know in what other situations this direct fitting approach might fail beyond the ones you’ve already discussed. Are there additional examples or patterns you noticed where the relationship breaks down?
3. Knits: L182 & L187 “BSNL” should be corrected to “BNSL.”
4. References that are relevant and also can be discussed in the paper:
   - Li et al. (2025) - “(Mis)Fitting: A Survey of Scaling Laws”
   - Mayilvahanan et al. (2025) - “LLMs on the Line: Data Determines Loss-to-Loss Scaling Laws”

---

> ### Author Response · Authors · 2025-11-17
>
> We thank the reviewer for the detailed and encouraging review. We appreciate the recognition of clear presentation, careful comparison with prior work and strong consistency across scales. Below we address the reviewer’s questions in detail.
>
> We have updated the submitted article to address the discussed questions. For the reviewer’s convenience, all changes in the updated paper are marked by using **green font color**.
>
> **Predicting the average across benchmarks.** We thank the reviewer for the suggestion to consider this approach and believe that including it in the paper will make our study more complete. We propose an approach to model the average across metrics in **Appendix G** of the updated submission. We initially take steps to ensure the single aggregate is meaningful. For metrics, where the scores are not by default in the range [0, 1] (i.e., multiple-choice tasks), we first normalize the accuracy as described in section 3.1 (we apply the transformation $Q' := (Q - Q_{\text{random}}) / (1 - Q_{\text{random}})$). We also exclude models where any of the benchmarks achieved a score of less than 5% points above the random performance to reduce noise.
>
> We note that it is not immediately obvious which functional form should be used to model the relationship between average score and training cost, as we are calculating the mean of separate, independent tasks. In our analysis, we take the empirical approach and examine whether the functional form of Equation 2 can be used in such a case. We exclude all models with more than 6e21 training FLOPs to validate the extrapolation and fit coefficients on the remaining experiments. The results are visualized in Figure 11, while error rates are outlined in Table 11. We observe good fit and extrapolation quality, with the validation MAE and MRE equal to 0.0116 and 2.08%, respectively.
>
> This initial examination confirms the possibility of using our approach for modeling and forecasting the averaged accuracy across tasks.
>
> **Benchmarks that are hard to predict.**  In our study, we focus on a suite of popular benchmarks commonly used to evaluate language models. These tasks are generally reliable for tracking improvements in model quality. However, it is not guaranteed that every downstream task will exhibit such smooth behaviour. There are several possible causes of unreliability. For example, in [1] the authors report “step function” behaviour in the scaling of BIG-Bench: CS Algorithms. This benchmark consists of two sub-tasks: *balanced parentheses* and *longest common subsequence*. The observed “breakthrough” and plateau around 0.5 may appear when the model first solves one of the sub-tasks. Another case of unpredictability arises when a metric is adversarially constructed to highlight specific difficulties in scaling language models, as in [2]. Finally, many downstream tasks exhibit emergent behaviour, where improvements are only visible after crossing a minimum training budget [3][4]. Therefore, we might form an observation that some metrics are unreliable if we measure their performance below this critical threshold.
>
> **Other comments.** We fixed the typos highlighted by the reviewer and updated the Related Work section with the provided references. We thank the reviewer for pointing out these issues.
>
> [1] Gadre et al., Language models scale reliably with over-training and on downstream tasks
>
> [2] Wei et al., Inverse scaling can become U-shaped
>
> [3] Wei et al., Emergent Abilities of Large Language Models
>
> [4] Du et al., Understanding Emergent Abilities of Language Models from the Loss Perspective

---

> > ### Author Response · Authors · 2025-11-25
> >
> > Dear Reviewer 5erS,
> >
> > We appreciate the thoughtful and encouraging review. If there are any clarifications or additional details that could help improve our submission, we would be happy to address them. Thank you!
> >
> > Authors

---

> > > ### Comment · Reviewer_5erS · 2025-11-26
> > >
> > > Thank you for the response. I don’t have further clarifications and will maintain my score.

---

### Author Response · Authors · 2025-12-02
**Rebuttal Summary**

Dear AC,


This year’s policy changes introduced an additional load of work to the Area Chairs. Therefore, for convenience, in this comment we summarize our responses and changes provided in the updated submission. During the discussion period, we have updated the paper and supplementary materials. All modified parts of the article are marked using **green font color**.

* Reviewer **5erS**:
   * The reviewer noted the clear writing, good fit of the metrics across scales and careful comparison with prior work.
   * Questions and weaknesses mentioned in the review included: 1) predicting a single aggregated metric, like the average accuracy; 2) Question about cases when fitting is difficult; 3) Typos; 4) Request for additional references.
   * To address the review we added: 1) **Appendix G** where we fit the average benchmark accuracy; 2) Explanation on the cases when predictability is complicated; 3) Fixed the typos in the text; 4) Added the references to the related work.
   * The reviewer sent a rebuttal reply specifying that no further clarifications were needed and **kept their positive evaluation (score 8)**.
* Reviewer **MNqJ**:
   * The reviewer noted wide coverage of benchmarks and careful comparison with prior work.
   * Questions and weaknesses mentioned in the review included: 1) Modeling the accuracy with irreducible error (for example, when some questions are mislabeled); 2) Question about motivation.
   * To address the review we added: 1) **Appendix H** where we propose and fit a version of the scaling law incorporating maximum achievable accuracy 2) Explanation of the motivation of our study.
   * We did not receive a reply from the reviewer during the rebuttal period; **the positive evaluation (score 6) was not changed**.
* Reviewer **yipV**:
   * The reviewer noted compatibility of the scaling law with multiple metrics, validation on held-out points, and comparison with the baseline methods.
   * Questions and weaknesses mentioned in the review included: 1) Concerns on the availability of code and data; 2) Concern about the lack of the details in hyperparameter tuning; 3) Predicting the average score across benchmarks; 4) Possibility of modeling the maximum achievable accuracy to account for ambiguous or mislabeled questions; 5) Question about the proxy metric used in the two-stage approach.
   * To address the review we added: 1) Clarification on reproducibility: Our training was based on an open-source repository detailed in **Section 3**. We also noted that we trained all models only on publicly available datasets (DCLM with Stackv2 and OpenMath Reasoning or C4). The details of the dataset mixture are available in **Section 3**. We additionally updated the supplementary material of the submission to include csv file with losses and benchmark scores of all models and repository with scaling law fitting code; 2) Details of tuning learning rate and batch size and updated Appendix F of the submission; 3) **Appendix G** with fit of the average score; 4) **Appendix H** with the fit of the scaling law with the inclusion of the irreducible error; 5) Clarification on the metric used in two-stage approach.
   * The reviewer replied that their concerns were addressed and **raised the score to 6**.


During the rebuttal period, we have engaged in the discussion and improved the submission based on the received feedback. We believe that the reviewer’s concerns are properly addressed. We appreciate the additional time taken to review our submission. Thank you!

---

### Meta-Review · Area_Chair_dnuv · 2025-12-28

**Summary:**

Reviewers had questions about motivation (why predict downstream accuracy instead of loss?) and some technical choices (e.g., how to model cases where there is some irreducible error arising from mislabeled examples). The authors did a good job responding to these questions in their rebuttal. In my opinion, no serious concerns remain. This is a solid paper that is well-written and proposes a simple method that can clearly impact practice.

**Reviewer Concerns:**

See main comment.

**Reviewer Scores:**

468 -> 668

---

### Decision · Program_Chairs · 2026-01-26

Accept (Poster)